# Combinatorial *GxGxE* CRISPR screen identifies SLC25A39 in mitochondrial glutathione transport linking iron homeostasis to OXPHOS

Xiaojian Shi [1,2,6], Bryn Reinstadler [3,4,5,6], Hardik Shah [3,4,5], Tsz-Leung To[3,4,5], Katie Byrne[1,2], Luanna Summer[1,2], Sarah E. Calvo[3,4,5], Olga Goldberger[3,4], John G. Doench [5], Vamsi K. Mootha [3,4,5] & Hongying Shen [1,2 ✉]

The SLC25 carrier family consists of 53 transporters that shuttle nutrients and co-factors across mitochondrial membranes. The family is highly redundant and their transport activities coupled to metabolic state. Here, we use a pooled, dual CRISPR screening strategy that knocks out pairs of transporters in four metabolic states — glucose, galactose, OXPHOS inhibition, and absence of pyruvate — designed to unmask the inter-dependence of these genes. In total, we screen 63 genes in four metabolic states, corresponding to 2016 single and pair-wise genetic perturbations. We recover 19 gene-by-environment (GxE) interactions and 9 gene-by-gene (GxG) interactions. One GxE interaction hit illustrates that the fitness defect in the mitochondrial folate carrier (SLC25A32) KO cells is genetically buffered in galactose due to a lack of substrate in de novo purine biosynthesis. GxG analysis highlights a buffering interaction between the iron transporter SLC25A37 (A37) and the poorly characterized SLC25A39 (A39). Mitochondrial metabolite profiling, organelle transport assays, and structure-guided mutagenesis identify A39 as critical for mitochondrial glutathione (GSH) import. Functional studies reveal that A39-mediated glutathione homeostasis and A37-mediated mitochondrial iron uptake operate jointly to support mitochondrial OXPHOS. Our work underscores the value of studying family-wide genetic interactions across different metabolic environments.

[1] Cellular and Molecular Physiology Department, Yale School of Medicine, New Haven, CT, USA. [2] Systems Biology Institute, Yale West Campus, West Haven, CT, USA. [3] Howard Hughes Medical Institute and Department of Molecular Biology, Massachusetts General Hospital, Boston, MA, USA. [4] Department of Systems Biology, Harvard Medical School, Boston, MA, USA. [5] Broad Institute, Cambridge, MA, USA. [6] These authors contributed equally: Xiaojian Shi, Bryn Reinstadler. ✉email: hongying.shen@yale.edu

The SLC25 family transporters play a critical role in shuttling metabolites into and out of mitochondria[1–3]. The protein family is evolutionarily conserved in nearly all eukaryotes, only missing in a few, highly diverged eukaryotes such as *Giardia lamblia* and *Encephalitozoon cuniculi*[4]. Among the 53 SLC25 proteins in human, the most well-characterized ones include adenine nucleotide carriers (ANTs) that exchange ADP for ATP to support cellular bioenergetics and UCP1 that is exclusively expressed in brown adipose tissue and dissipates proton motive force to generate heat. Despite the critical roles, approximately 20 members remain transporters of unknown functions.

Elegant structural characterization of the well-known ANT enables structure-guided investigations on the SLC25 transporters of unknown functions. Each SLC25 transporter is a structural and functional monomer of approximately 300 amino acids with six transmembrane α-helices (3-fold pseudo-symmetrical repeats of the two transmembrane helices) surrounding the central cavity. The ANT crystal structures in both cytoplasmic-open (c-state) and matrix-open states (m-state) locked by two inhibitors bound at the solute binding site, carboxyatractyloside and bongkrekic acid respectively, highlighted a conserved alternating-access transport mechanism[5] enabled by the conformational change upon solute binding at the central cavity[6,7]. The high-level sequence homology and structural similarity among the SLC25 transporters enable both identification of solute-binding residues and prediction of the transporting solute based on the putative chemical groups.

A major challenge for the systematic interrogation of SLC25 transporter functions and their metabolic regulations lies in the high level of redundancy within the family. For instance, 36 members are part of paralogous subgroups[8], including four ANT (exchanging ATP for ADP) and four calcium-binding mitochondrial carrier proteins SCaMCs (enabling net adenylate transport). Previously, in our genome-wide CRISPR screen in galactose, a low glucose condition designed to discover genes required for mitochondrial OXPHOS, none of the four ANTs scored as hits[9]. Yet, the ANT inhibitor, bongkrekic acid, induces cell death in low glucose[10].

Now, inspired by the mitochondria-focused genetic interaction map in *Saccharomyces cerevisiae*[11] and the known influence of environment on genetic interactions[12], we sought a dual genetic perturbation approach to probe the human SLC25 transporters under different metabolic states[13]. We applied a combinatorial CRISPR screening approach to interrogate SLC25 transporters in cellular physiology. By screening in four different media conditions, we uncovered their metabolic regulation, including the fitness defects in the mitochondrial folate transporter SLC25A32 KO cells that can be buffered in galactose condition due to substrate limitation in de novo purine biosynthesis. Amongst transporters of unknown function, our screen highlighted SLC25A39, in which SLC25A39 KO cells' growth defect is buffered by antimycin or by KO of the mitochondrial iron transporter SLC25A37. Follow-up studies combining mitochondrial metabolite profiling, organelle transport assay and structure-guided mutagenesis identified a critical role of SLC25A39 in mitochondrial glutathione uptake, a discovery that has also been independently reached by a recent report[14]. The buffering genetic interaction revealed from our CRISPR screen[13] allowed further extending the finding by demonstrating a functional coordination of mitochondrial glutathione uptake (mediated by SLC25A39) and mitochondrial iron import (mediated by SLC25A37) in supporting OXPHOS.

## Results

**Combinatorial CRISPR screening approach to interrogate SLC25 transporters**. We utilized a dual Cas9 enzyme-based

knockout strategy[15] to probe all 53 human SLC25 family members in a pair-wise manner in pooled format. This dual Cas9 system takes advantage of different protospacer adjacent motif (PAM) sequences recognized by *Streptococcus pyogenes* Cas9 (SpCas9) versus *Staphylococcus aureus* Cas9 (SaCas9), enabling simultaneously knocking out two genes (Fig. 1a). To construct the pPapi-SLC25 library, SpCas9 and SaCas9 guides were designed to target 53 SLC25 carriers, seven additional SLC proteins present in MitoCarta 2.0[16], BCL2L1, MCL1 (the two paralogous anti-apoptotic proteins) and EEF2 (elongation factor 2, essential for protein translation) as positive controls, 8 non-cutting controls and 19 cutting controls that target olfactory receptors (ORs) as negative controls. This leads to a custom pPapi-SLC25 library of 273×273 = 74,529 plasmids for the pooled screen (Fig. 1b).

We carried out a growth fitness screen in chronic myelogenous leukemia K562 cells across four media conditions (Fig. 1a): (1) standard high glucose media (25 mM glucose, glc), (2) galactose media in which the glucose is replaced by the same concentration of galactose (25 mM galactose, gal), (3) full media with 100 nM antimycin (antimycin), and (4) full glucose media but lacking pyruvate (−pyruvate). High glucose condition supports cellular bioenergetics using both glycolysis and OXPHOS. Galactose is an epimer of glucose, which only permits mitochondrial OXPHOS ATP production, due to slower kinetics to utilize galactose in glycolysis[17]. Antimycin is a known inhibitor of the mitochondrial electron transport chain (ETC) complex III inhibiting OXPHOS. Removing pyruvate in the media prevents the recycling of NADH into NAD + by the cytosolic LDH enzyme, forcing mitochondrial redox shuttle systems to support cytosolic redox balance[18]. All four media conditions are supplemented with uridine and pyruvate (except no pyruvate in –pyruvate condition), which are required for the cellular growth under OXPHOS inhibition[19]. In the screen, the cells grew exponentially with expected growth rates: glucose ~ −pyruvate > antimycin > galactose (Fig. 1c). The positive control pair BCL2L1 x MCL1 exhibited the expected synthetic sick genetic interaction as previously reported across all conditions (Fig. 1d)[15].

We quantified fitness in each condition as the log2-fold-change (LFC) of sgRNA abundance at fifteen days over plasmid DNA (Supplementary Data 1). After applying quality control filters (Supplementary Fig. 1 and Methods), we combined the data from two screening batches, and combined SaCas9 and SpCas9 targeting guides for the same genes. It is worth noting that we used cutting guides as the negative controls to also control for the deleterious effect during the double-stranded DNA cutting[20] (see Methods). Together, we scored the genetic perturbation for 63 genes, giving a total of ($63^2 – ½ (63^2 – 63) =$) 2016 genetic perturbations, or a total of (2016 ×4 =) 8064 gene by environment conditions.

We then proceeded to identify three categories of interactions: (i) GxE (Gene x Environment) interaction, wherein a single SLC25 gene KO differentially affects cellular fitness under certain metabolic states; (ii) GxG (Gene x Gene) interaction, wherein the loss of two genes leads to a fitness phenotype that is not the simple additive sum of two single KOs, and (iii) GxGxE (Gene x Gene x Environment) interaction, wherein the impact of the pairwise genetic interaction is dependent on the metabolic state.

The results immediately revealed the value of screening pairs of knockouts across different metabolic conditions. Of 53 SLC25 family members, 6 genes scored significantly as essential in at least one of the four media conditions. They include SLC25A1 (citrate transporter), SLC25A26 (SAM transporter) and MTCH2 (unknown) in glucose; SLC25A1, SLC25A3 (phosphate transporter), SLC25A19 (thiamine pyrophosphoate transporter) and SLC25A26 in galactose; SLC25A1, SLC25A3 and SLC25A26 in

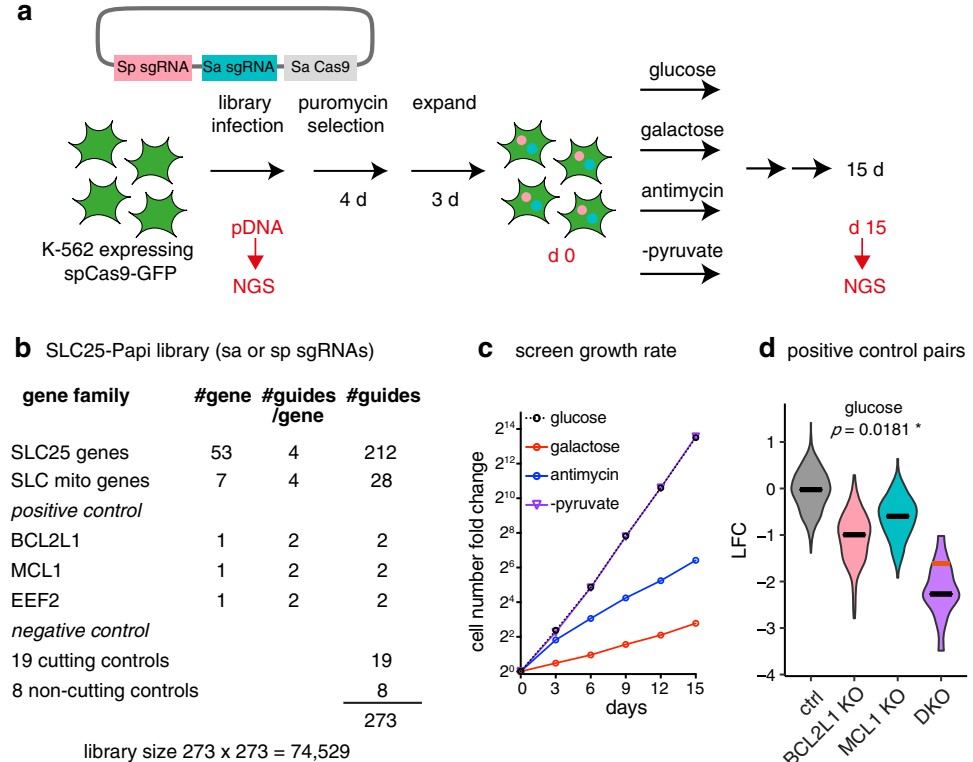

**Fig. 1 Combinatorial CRISPR screen of the SLC25 family transporters across four media conditions. a** Schematic overview of the combinatorial CRISPR strategy. **b** SLC25-Papi library construction. **c** Growth rate of the pooled CRISPR screen. The batch 2 growth rate was shown and representative of the two batches. **d** positive control pair BCL2L1 and MCL1. Data shown are log2 fold changes from day 15 in the glucose condition. Gray, control-control pair; pink, BCL2L1 x Ctrl or Ctrl x BCL2L1; blue, MCL1 x Ctrl or Ctrl x MCL1; and purple, the double knockouts of BCL2L1 x MCL1 or MCL1xBCL2L1. The black lines indicate the median and the red line indicates the expected value for the double knockout based on the sum of the single knockouts when there is no genetic interaction. Statistical significance was calculated using two-tailed *t* test. Source data are provided as a Source Data file.

antimycin; SLC25A1, SLC25A19, SLC25A26, SLC25A32 (folate transporter) and MTCH2 in -pyruvate condition. Importantly, directly comparing the single KO phenotype across four conditions highlighted 19 nutrient-dependent phenotypes that are otherwise mild in any single condition, which we termed GxE interactions (Fig. 2a and Supplementary Data 2). In glucose, 5 pairs of genes exhibited a pairwise GxG interaction, and 9 GxG interactions scored in at least one of the four conditions, including 8 GxG interactions that are condition-specific (Fig. 3a and see below). These observations indicate that SLC25 transporters exhibit considerable functional redundancy and that phenotypes can be unmasked only by combinatorial screening under different conditions.

**Gene x environment (GxE) interactions**. We quantified the resulting GxE interaction effects by calculating the Z-score of the LFC (relative to pDNA) using cutting control guides as the null distribution, which corrects for differences in growth rates between different conditions (Supplementary Data 2). As mentioned above, a total of 19 metabolic state-dependent phenotypes (GxE hits) were discovered (Fig. 2a and Supplementary Data 2). We followed up on two hits that exhibited the strongest GxE interactions, SLC25A19 and SLC25A32, which are transporters of two B family vitamin-derived cofactors, thiamine pyrophosphate (ThPP, or vitamin B$_1$) and folate species (vitamin B$_9$), respectively.

**GxE interactions for ThPP carrier SLC25A19**. Loss of SLC25A19 results in severe growth defects in glucose, galactose and -pyruvate conditions, which is buffered in the antimycin

condition when the respiratory chain is inhibited (Supplementary Fig. 2a). SLC25A19 transports ThPP[21], the critical cofactor for the E1 subunit of the mitochondrial pyruvate dehydrogenase complex (PDC) and oxoglutarate dehydrogenase complex (OGDC) (Supplementary Fig. 2b). Mutations in SLC25A19 cause Amish lethal microcephaly (MCPHA) and thiamine metabolism dysfunction syndrome 4. We first validated the buffering phenotype in the follow-up studies, in which SLC25A19 exhibited milder growth fitness defect compared to the control cells in the presence of complex III inhibitor antimycin, complex I inhibitor piericidin and complex V inhibitor oligomycin than in the regular culture condition (Supplementary Fig. 2c–d). Consistently, our recent genome-wide CRISPR screens performed in the presence of mitochondrial inhibitors also identified that cell fitness defect in the SLC25A19 KOs can be buffered by a spectrum of mitochondrial inhibitors (Supplementary Fig. 2e)[22], suggesting a direct impairment of OXPHOS function by SLC25A19 loss.

**GxE interactions for the mitochondrial folate carrier SLC25A32**. The fitness defects of SLC25A32 KOs observed in glucose, antimycin and −pyruvate conditions are buffered in galactose in the screen (Fig. 2a, b). We first confirmed the screen result in the follow-up experiments and showed that two lines of SLC25A32 CRISPR KO cells proliferated slower than the control, but the growth rate of these KOs were comparable to the control cells in the galactose condition (Fig. 2c and Supplementary Fig. 2f).

SLC25A32 is the mitochondrial carrier for the tetrahydrofolate (THF) species in mitochondrial one-carbon (1 C) metabolism, as demonstrated by in vitro studies[23], mouse genetics[24] and cell

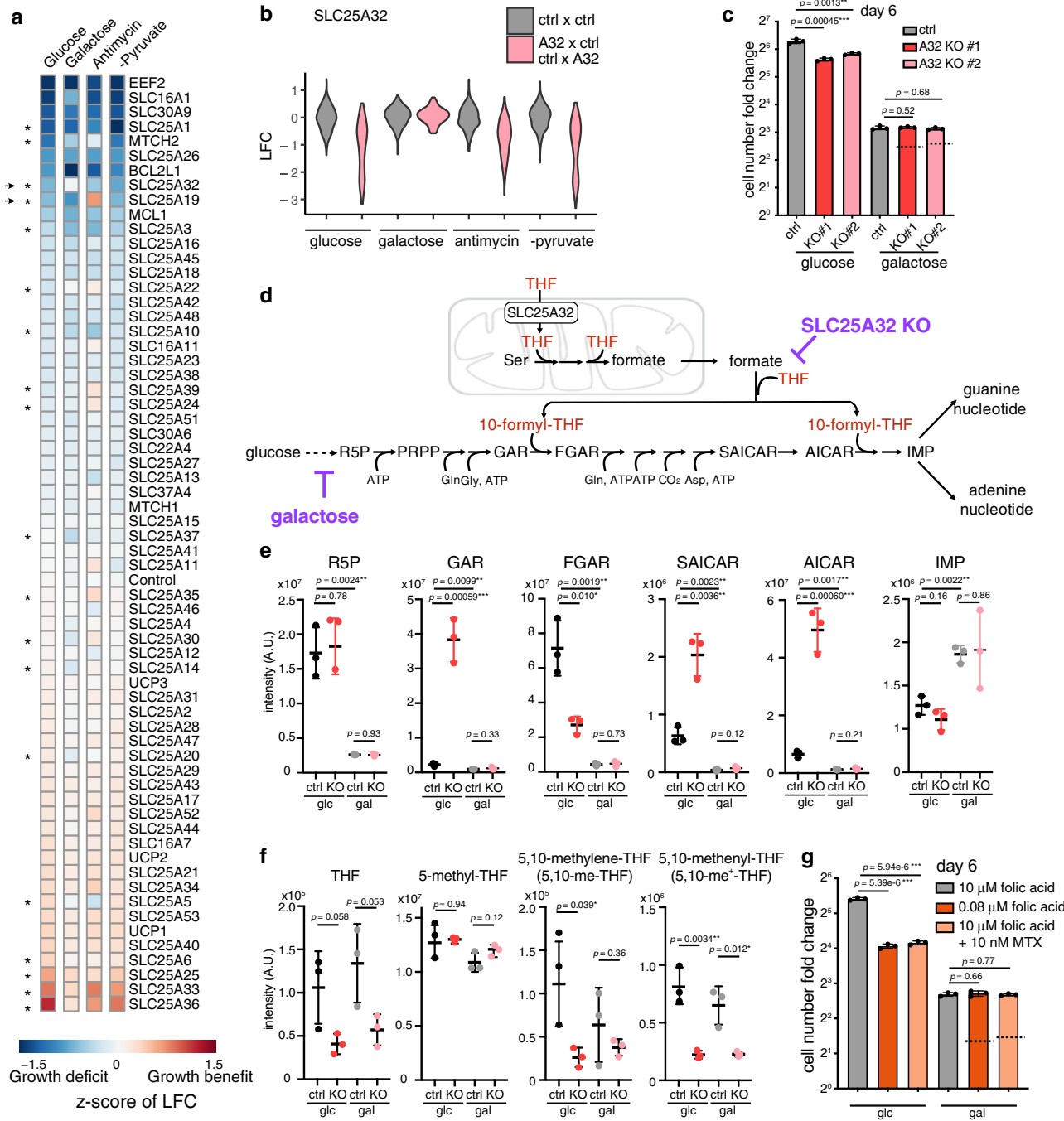

**Fig. 2 Gene x Environment (GxE) mapping indicates that loss of mitochondrial folate metabolism can be buffered in galactose condition. a** A heatmap depicts the fitness of single gene KOs across all four media conditions. Values are Z-scores of log fold change (LFC) value of the targeting guide abundance. Asterisks highlight the 19 significant "GxE" interactions, i.e., those knockouts that cause fitness phenotypes differentially across the conditions. Arrows indicate two hits that were experimentally validated and pursued in the current study: SLC25A32 and SLC25A19 (see Supplementary Fig 2). **b** SLC25A32 single KO phenotype across four conditions in the screen, showing the buffering effect in galactose. Gray, the control-control distribution; pink, SLC25A32 single KOs ("A32 x Ctrl" or "Ctrl x A32" combined). **c** Follow-up studies show growth fitness of SLC25A32 CRISPR KO cells generated by two different sgRNAs in glucose and galactose conditions. The dotted line denotes the expected LFC, based on an additive model, if there was no genetic interaction ($n = 3$). **d** de novo purine biosynthesis pathway. **e** LC-MS measurement of purine intermediates in SLC25A32 KO and control cells in glucose and galactose condition for 2 days ($n = 3$). **f** LC-MS measurement of tetrahydrofolate (THF) cofactors in SLC25A32 KO and control cells in glucose and galactose condition for 2 days ($n = 3$). **g** Growth fitness of wild type cells either cultured in low folate media (0.08 μM) or treated with DHFR inhibitor methotrexate at sub-IC50 dose (10 nM), in glucose and galactose condition. The dotted line denotes the expected LFC, based on an additive model, if there was no genetic interaction ($n = 3$). Statistical significance was calculated using two-tailed $t$ test. Significance level were indicated as *** $p < 0.001$, ** $p < 0.01$, * $p < 0.05$ and n.s. $p > 0.05$. Data are expressed as mean ± SD. Source data are provided as a Source Data file.

culture characterization[25–27]. SLC25A32 was also postulated to transport cofactor FAD[28], and its mutations in humans cause riboflavin-responsive exercise intolerance[29,30]. As FAD is the cofactor of the ETC complex II and fatty acid oxidation critical for mitochondrial bioenergetics, a lack of fitness defect of the SLC25A32 KOs in galactose when cells depend on respiration for survival (Fig. 2c and Supplementary Fig. 2f) is surprising, and indicates that either SLC25A32 does not transport FAD into mitochondria in K562 cells, or perhaps that redundant mechanisms exist.

We hypothesized that the observed buffering interaction between the galactose condition and the SLC25A32 KO is due to their perturbations converge on the de novo purine biosynthesis. SLC25A32 transports THF into mitochondria to support one-carbon (1 C) metabolism that converts serine into formate, which is then exported into the cytosol for de novo purine biosynthesis (Fig. 2d). In the absence of SLC25A32, cells dramatically accumulated purine intermediates, GAR and AICAR, the substrates of de novo purine biosynthesis enzymes utilizing 10-formyl-THF (Fig. 2e), suggesting perturbation in the pathway. On the other hand, glucose metabolism through pentose phosphate pathway (PPP) generates ribose-5-phosphate (R5P) providing the initial substrate for de novo purine biosynthesis. Replacing glucose with galactose severely depletes de novo purine biosynthesis pathway intermediates in both control and KO cells (Fig. 2e). Folate species measurement corroborates these results: consistent with a role of SLC25A32 in supporting mitochondrial 1 C metabolism, SLC25A32 KOs have significantly reduced cellular 5,10- me$^+$-THF (Fig. 2f), the mitochondrial folate cofactor that is generated by redox-mediated MTHFD2 (Supplementary Fig. 2g). Interestingly, the KO cells also have slightly, albeit not significantly, reduced THF and 5,10-me-THF (Fig. 2f), which may suggest a primary mitochondrial localization of these folate species.

The long-term galactose adaptation might directly inhibit THF cofactor biosynthesis, providing additional means to buffer the defective mitochondrial THF import in the SLC25A32 KOs. Two enzymes involved in cytosolic THF biosynthesis, DHFR and MTHFR, utilize NADPH cofactors (Supplementary Fig. 2g). Long-term galactose adaptation (2 weeks) appeared to reduce the cellular NADH/NAD$^+$ ratio and NADPH/NADP$^+$ ratio, albeit not significantly (Supplementary Fig. 2i), as well as all major THF species (Supplementary Fig. 2h). The data are consistent with the report that decreased PPP flux and G6PD activity might impair NADPH and THF biosynthesis[31]. The exception is an increase of the mitochondrial 5,10-me$^+$-THF, likely due to an upregulation of the mitochondrial 1 C metabolism in galactose[32].

We further extended the observed buffering interaction by directly inhibiting the folate pathway with either low-folate media or a well-known folate pathway inhibitor, methotrexate. Cells cultured in low folate or treated with sub-IC50 concentration of methotrexate proliferated slower than the controls, and this fitness defect is completely blunted while culturing cells in galactose (Fig. 2g). Here, by converting the GxE interaction to an environment-by-environment (ExE) interaction, we show that PPP flux might modulate folate metabolism. In particular, a lower ribose biosynthesis rate might mask folate deficiency or confer methotrexate resistance, with potentially broad implications in metabolic disorders and cancer therapy.

**GxGxE interactions in paralogous genes**. To explore pairwise genetic interactions, we scored the strength of the GxG interactions using the π-score[33] (Supplementary Fig. 3a), which allows comparison of the fitness consequences of a double knockout relative to what is expected by the simple addition of two single

knockouts. For genetic perturbations that confer fitness defects, a negative π-score defines "synthetic sick" and a positive π-score defines "buffering", respectively (Supplementary Data 3). Scores for all GxG pairs are summarized in Supplementary Figs. 6–9. Here, we focus on SLC25 gene pairs with large effect sizes (absolute π-score >0.25) and with significance (FDR < 2%). We excluded gene pairs exhibiting "ceiling" and "deader than dead" effects (see Methods).

This procedure for finding genetic interactions highlighted nine SLC25 gene pairs of genetic interaction that scored in at least one media condition (Fig. 3a and Supplementary Data 4). We validated these interactions in the follow-up studies, including A5 x A6 (galactose, Fig.3c), A37 x A39 (glucose, antimycin, Fig. 3g), A28 x A37 (glucose and galactose, Supplementary Fig. 3d), A36 x MTCH2 (glucose, Supplementary Fig. 3e), A1 x A36 (glucose, Supplementary Fig. 3f), MTCH1 x MTCH2 (glucose, antimycin and galactose, Supplementary Fig. 3g), A20 x MTCH2 (antimycin, Supplementary Fig. 3h). The positive control GxG pair (BCL2L x MCL1) and the negative control pair for SLC25 genes that are not expressed (UCP1 and SLC25A31) are shown at the bottom for comparison (Fig. 3a, Supplementary Note 1 and Supplementary Data 5).

Supporting the notion that SLC25 proteins are highly redundant, we observed strong synthetic sick interactions between 4 paralogous gene pairs:[8] SLC25A5 x SLC25A6 (ANTs), MTCH1 x MTCH2 (two outer mitochondrial membrane SLC25 proteins of unknown functions involved in mitochondria dynamics[34,35,36]), SLC25A16 x SLC25A42 (CoA carriers)[37,38], and SLC25A28 x SLC25A37 (iron carriers)[39].

Among these paralogs, two ANTs, SLC25A5 (ANT2) and SLC25A6 (ANT3), exhibit the strongest synthetic sick interaction in galactose (Fig. 3c, Supplementary Data 4) and the greatest difference in genetic interactions across conditions (Fig. 3a). ANTs typically import ADP$^{3-}$ in exchange for OXPHOS-produced ATP$^{4-}$ across the inner mitochondrial membrane (Fig. 3d). The human genome encodes four ANTs, and K562 cells express higher level of SLC25A5 (ANT2) and SLC25A6 (ANT3) than SLC25A4 (ANT1), while SLC25A31 (ANT4) is not expressed (Fig. 3e). SLC25A4 is not upregulated in the SLC25A5 and SLC25A6 DKOs (Fig. 3f), and SLC25A31 cannot be detected in any single or DKOs. In glucose, SLC25A5 × SLC25A6 DKOs exhibit no fitness defect (Supplementary Fig. 3b), however, the same DKO cells exhibited an extreme synthetic sick phenotype in galactose (Fig. 3b, c), suggesting a redundant but critical role of SLC25A5 and SLC25A6 in supporting OXPHOS when glycolysis is severely inhibited. This redundancy helps to explain why neither carrier scored in our previous "Glu/Gal" screen[9].

**SLC25A39 KO exhibits both GxG and GxE interactions**. Our screen recovered GxG interactions between five non-paralogous gene pairs (Fig. 3a) and we focused on a poorly characterized transporter SLC25A39 (A39), which exhibits a buffering interaction with the iron transporter SLC25A37 in all four conditions. A stronger interaction occurred in the presence of antimycin (Supplementary Fig. 3c), which we validated in the follow-up experiments (Fig. 3g). A39 was of additional interest as it also scored as a GxE screening hit in which the growth fitness defect in the A39 KO cells was buffered in antimycin (Fig. 2a and Supplementary Fig. 3c).

SLC25A39 is a transporter of unknown function. A39 mutant flies show defects in neurotransmission leading to degeneration of the eyes[40]. Knockdown of A39 in zebrafish leads to profound anemia[41]. Yeast mutant of A39 ortholog $mtm1\Delta$ exhibited altered mitochondrial iron homeostasis that could be complemented by the vertebrate A39 sequence[41,42]. A39 resides in a susceptibility

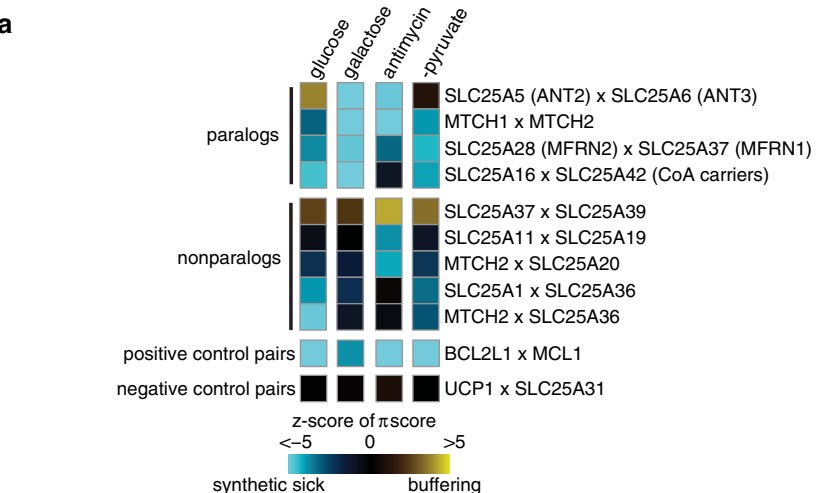

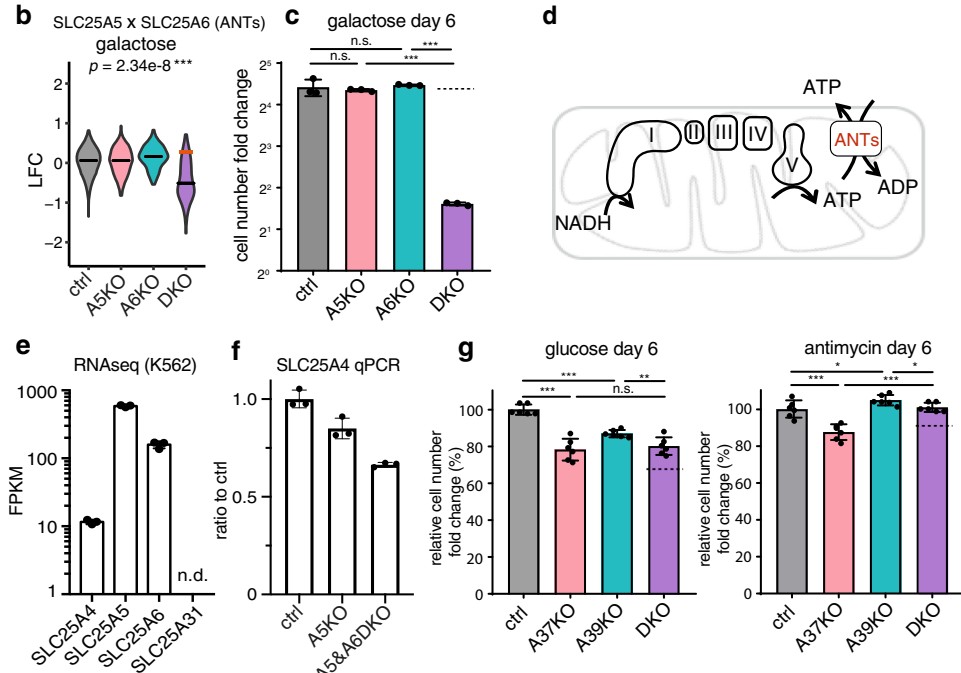

**Fig. 3 Identification of Gene x Gene x Environment (GxGxE) interactions. a** Heatmap shows the π-score for pairwise genetic interaction. The values are z-transformed to compare across media conditions, as described in the methods. For comparison, a positive control pair (BCL2L1 x MCL1) that are always detrimental when jointly knocked out and another pair (UCP1 x SLC25A31) that do not interact are shown. The rows and columns are not clustered. **b** LFC values from the CRISPR screen are shown to illustrate the strongest synthetic sick genetic interaction observed from the CRISPR screen, SLC25A5 and SLC25A6 loss in the galactose condition. The red line indicates the expected LFC, based on an additive model, if there were no genetic interaction. The black line indicates the actual median LFC value. **c** Follow-up validation of the genetic interaction between SLC25A5 and SLC25A6 in the galactose ($p_{ctrl/A5KO} = 0.59$, $p_{ctrl/A6KO} = 0.66$, $p_{A5KO/DKO} = 2.11 \times 10^{-7}$, $p_{A6KO/DKO} = 1.93 \times 10^{-8}$). The dotted line indicates the expected growth fitness, based on an additive model, if there were no specific genetic interaction ($n = 3$). **d** The cartoon illustrates the function of ANTs in supporting bioenergetics. **e** Transcript levels of the four ANTs in K562 cells (data plotted from the published PBS vehicle treated K562 cells RNA-seq dataset GSE74999) ($n = 3$). **f** qPCR of SLC25A4 mRNA level in SLC25A5 KO and SLC25A5 and SLC25A6 DKO cells ($n = 3$). **g** Follow-up validation of the genetic interaction between SLC25A37 and SLC25A39 in the glucose ($p_{ctrl/A37KO} = 1.04 \times 10^{-5}$, $p_{ctrl/A39KO} = 3.08 \times 10^{-6}$, $p_{A37KO/DKO} = 0.56$, $p_{A39KO/DKO} = 0.0092$) and antimycin ($p_{ctrl/A37KO} = 0.00074$, $p_{ctrl/A39KO} = 0.0498$, $p_{A37KO/DKO} = 5.54 \times 10^{-5}$, $p_{A39KO/DKO} = 0.030$) conditions ($n = 6$). Statistical significance was calculated using two-tailed $t$ test. Significance level were indicated as *** $p < 0.001$, ** $p < 0.01$, * $p < 0.05$ and n.s. $p > 0.05$. Data are expressed as mean ± SD. Source data are provided as a Source Data file.

locus for epilepsy[43], and de novo frameshift variant[44] and damaging missense variants[45,46] are reported in autism spectrum disorder.

We sought to characterize A39's function by evaluating the impact on mitochondrial metabolism upon knocking out A39. A39 CRISPR KO cells exhibited a mild growth defect in glucose

(Supplementary Fig. 4a). To profile mitochondrial metabolism, we introduced into K562 cells the previously developed mitoIP construct (3xHA-eGFP-OMP25) and the control construct (3xMyc-eGFP-OMP25)[47,48]. Rapid immunopurification enabled a specific enrichment of mitochondria with little contamination from the ER or lysosome (Supplementary Fig. 4b). Metabolite

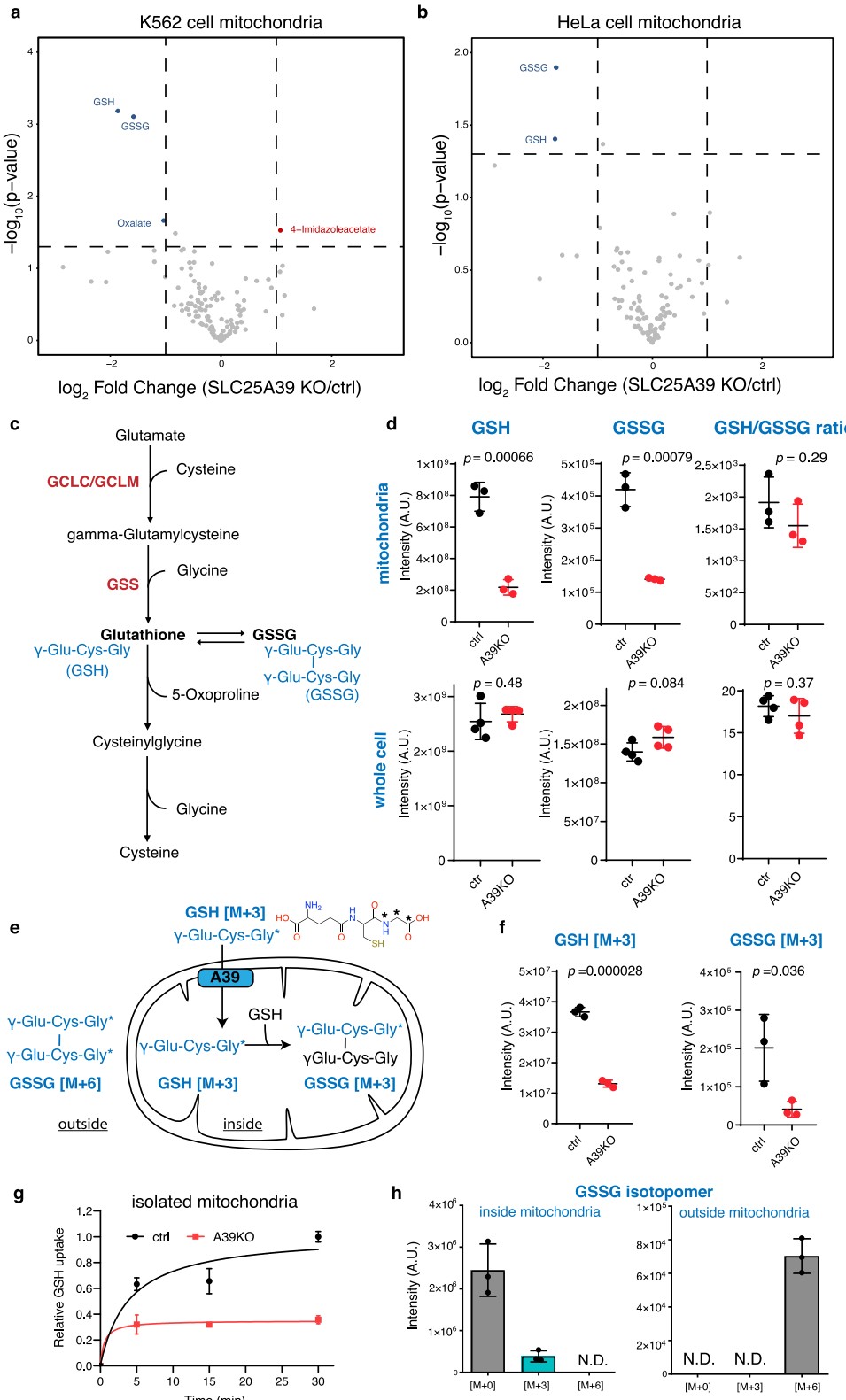

profiling of the isolated mitochondria revealed a striking depletion of both the reduced and the oxidized glutathione, GSH and GSSG, in the A39 KO mitochondria compared to the controls (Fig. 4a, d, and Supplementary Data 6), a result that is also replicated using an independent guide RNA against A39 (Supplementary Fig. 4d). We could also validate this finding in the A39 KO Hela cells (Fig. 4b, Supplementary Fig. 4c,and

Supplementary Data 7). The GSH/GSSG ratio was not changed (Fig. 4d). For both GSH and GSSG, we did not observe any changes in the whole cell extract (Fig. 4d), suggesting a specific regulation of the mitochondrial glutathione. Collectively, mitochondrial metabolite profiling shows a decrease of the reduced and oxidized forms of intramitochondrial glutathione upon A39 loss.

**Fig. 4 SLC25A39 is required for mitochondrial glutathione transport.** Volcano plots of mitochondrial metabolites from the A39 KO and control cells in K562 cells (**a**) and HeLa cells (**b**) highlighting a depletion of reduced and oxidized glutathione, GSH and GSSG, respectively, in the A39 KO mitochondria. **c** schematic of the biosynthesis and degradation of glutathione in mammalian cells. GSH is a tripeptide γ-Glu-Cys-Gly exclusively synthesized in the cytosol by two cytosolic enzyme systems, γ-glutamylcysteine synthetase (GCLC and GCLM) and GSH synthetase (GSS) **d** LC-MS measurement of reduced GSH, oxidized GSSG and GSH/GSSG ratio in isolated mitochondria ($n = 3$) and whole cells ($n = 4$). **e** Schematic overview of the organelle-based GSH uptake assay. Note that the imported labeled GSH can react with endogenous, unlabeled GSH to generate GSSG. **f** LC-MS measurement of mitochondrial labeled GSH and singly labeled GSSG in the uptake assay (30 min) ($n = 3$). **g** Time-course of labeled GSH uptake into mitochondria ($n = 3$). **h** GSSG isotopomer distribution from the inside (washed mitochondria fraction) and the outside (assay supernatant) of the control mitochondria in the GSH uptake assay at the 30 min time point ($n = 3$). Statistical significance was calculated using two-tailed $t$ test. Significance level were indicated as *** $p < 0.001$, ** $p < 0.01$, * $p < 0.05$ and n.s. $p > 0.05$. Data are expressed as mean ± SD. Source data are provided as a Source Data file.

We hypothesized that GSH is the solute transported across the mitochondrial membrane by A39. GSH is a tripeptide that is synthesized in the cytosol from three amino acids glutamate, cysteine and glycine (Fig. 4c). To test if GSH is transported in an A39-dependent manner, we designed an organelle-based GSH uptake assay. In the assay, we incubated isolated mitochondria from the control and A39 KO cells with 200 μM stable isotope-labeled GSH [M + 3] (GSH-[glycine-13C2, 15 N]), in which two carbons and one nitrogen of the glycine residue are labeled (Fig. 4e). After incubating at room temperature, we washed mitochondria and analyzed GSH level by LC-MS. The abundance of labeled GSH is significantly decreased in the KO mitochondria compared to the controls, confirming defective GSH uptake (Fig. 4f, g).

Stable isotope analysis of GSSG from this uptake assay further suggests that GSH is indeed imported into the mitochondria and functionally used to synthesize GSSG intramitochondrially. We analyzed the metabolites that are either outside (supernatant fraction) or inside mitochondria (washed mitochondrial fraction) from the control mitochondrial uptake assay samples at 30 min time point. In the outside fraction, we only detected doubly labeled GSSG, probably due to spontaneous oxidation of the labeled GSH, and no singly labeled GSSG was detected (Fig. 4h). However, in the inside fraction, a significant 13.4% of GSSG was detected as the singly labeled isotopomer [M + 3] (one GSH molecule is labeled), and no doubly labeled GSSG [M + 6] (both molecules of GSH are labeled) was detected (Fig. 4h). The most parsimonious explanation for the result is that labeled GSH is imported into the mitochondria and reacts with endogenous, unlabeled intramitochondrial GSH to generate singly labeled GSSG (Fig. 4e). The same GSSG isotopomer patterns were observed in the A39 KO mitochondria uptake samples (Supplementary Fig. 4e), suggesting a similar metabolic fate of the imported GSH in the KO mitochondria. Consistent with decreased labeled GSH in the KO mitochondria, the GSSG isomer [M + 3] was decreased in the A39 KO mitochondria (Fig. 4f). Together, the organelle-based GSH uptake assay and stable-isotope analysis suggest that A39 is required for mitochondrial GSH transport to support intramitochondrial conversion to GSSG.

We next used structural modeling and mutagenesis to gain additional evidence that A39 directly recognizes GSH. The atomic structures of ANTs in the c-state and m-state conformation[7] highlighted a putative solute binding site in the central cavity that is conserved across SLC25 transporters. Previous structure-function analysis on another amino acid ornithine carrier SLC25A15 (ORNT1) identified the critical RE residues (Fig. 5a) at this site for binding Cα-carboxylate and amino groups respectively, two signature chemical groups in all amino acids[49–51]. Supporting this binding specificity, we found that the corresponding residues were replaced by the KQ residues in the citrate carrier SLC25A1 (TXTP), which retains the positive charged Lys for binding carboxylate group but lacks the

negatively charged residue required for amino group binding. We then aligned eukaryotic A39 orthologs sequences spanning diverse taxa including protist *Plasmodium* species against human SLC25A15 (ORNT1) and SLC25A1 (TXTP) for comparision, and identified two corresponding, highly conserved Arg225 and Asp226 residues (Fig. 5a), supporting A39's amino acid binding. We then modeled the human A39 structure based on the ANT structure c-state conformation and confirmed that the Arg225 and Asp226 residues are indeed facing the central cavity allowing solute access (Fig. 5b). Because GSH contains a γ-amide bond preserving the carboxylate and amino group of the glutamate residue of GSH, we hypothesized that the RD residues are critical for A39's binding to the glutamate residue of GSH during its transport (Fig. 5b).

To test this solute recognition mechanism, we expressed either wild type A39[WT] or A39[D226A] mutant into the A39 KO cells. Western blotting the isolated mitochondrial lysates using an anti-A39 antibody demonstrated a depletion of A39 in the KO cells, and the tagged A39 proteins migrated at a slightly higher molecular weight (Fig. 5c and Supplementary Fig. 4g). The ectopically expressed wild type and mutant A39 proteins are expressed at a comparable level in the mitochondria, suggesting that the mutations do not affect protein stability (Fig. 5c). To explore the functional importance of A39's predicted GSH binding, we assay how this amino acid-binding defective mutant A39[D226A] impacts the mitochondrial GSH. Using mitochondrial metabolite profiling, the depleted mitochondrial GSH observed in the A39 KO cells could be rescued by re-expressing A39[WT], but not by the solute-binding defective mutant A39[D226A] (Fig. 5d). Consistently, the proliferation defect of the A39 KO cells can only be rescued by the wild type A39, but not by any solute binding defective mutants, A39[D226A] or A39[R225A/D226A] (Supplementary Fig. 4g and h). We also tested another A39[K329A] mutant in which a positively charged residue is highly conserved among all SLC25 transporters (Supplementary Fig. 4f), and the corresponding Arg residue in SLC25A15 (ONRT1) was found not directly involved in solute binding but in mediating solute-induced conformational changes critical for the transport activity[49]. Indeed, this A39[K329A] mutant failed to rescue the cell proliferation defect providing additional support that conformational change-mediated transport is central to A39 function. Interestingly, a high level of GSH at 10 mg/L in the culture media failed to rescue the fitness defect in the A39 KO cells (Supplementary Fig. 4i), suggesting that cellular GSH is not the limiting factor for mitochondrial GSH uptake and that is consistent with the observation that cellular GSH is not affected in the A39 KO cells (Fig. 4d). Collectively, these studies indicate that A39 is required for GSH transport into mitochondria.

We sought to validate and understand the mechanistic basis for the buffering interaction between A39 and A37 loss. We hypothesized that A39-mediated glutathione homeostasis converges onto A37-mediated mitochondrial iron uptake and together support mitochondrial OXPHOS. Notably, A39 is also

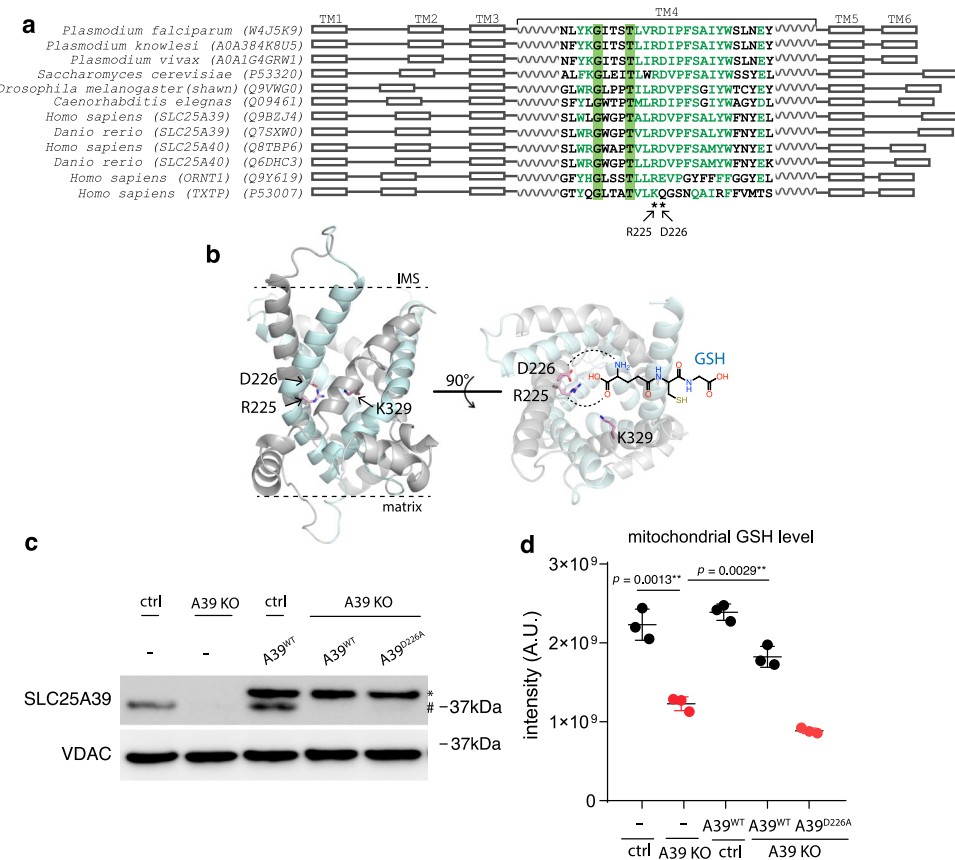

**Fig. 5 Structural modeling and functional mutagenesis of SLC25A39. a** Multiple sequence alignment of the eukaryotic A39 ortholog sequences spanning diverse taxa alongside with the human amino acid ornithine transporter SLC25A15 (ORNT1) and the human citrate transporter SLC25A1 (TXTP). The positions for the human SLC25A39 Arg225 and Asp226 residues predicted to bind amino acids are labeled by asterisks. **b** The modeled human A39 structure based on the ANT structure c-state conformation (PDB: 1OKC), in both side view (left) and cytoplasmic/mitochondrial intermembrane space (IMS) view (right). Odd transmembrane domain (TM1, 3, 5) are shown in gray and even (TM2, 4, 6) are shown in cyan. Two predicted GSH binding residues (R225, D226) and the predicted residue critical for solute-binding induced conformational change (K329) are shown. The predicted interaction between R225 and D226 with the carboxylate and amino group of the glutamate residue of GSH is shown by dashed line. **c** Western blotting showing the endogenous A39 (#) and ectopically expressed A39 protein (*). VDAC was used as the loading control. **d** LC-MS measurement of mitochondrial GSH level showing that the predicted substrate binding mutant A39$^{D226A}$ cannot restore mitochondrial GSH level in the A39 KO cells ($n = 3$). Statistical significance was calculated using two-tailed $t$ test. Significance level were indicated as *** $p < 0.001$, ** $p < 0.01$, * $p < 0.05$ and n.s. $p > 0.05$. Data are expressed as mean ± SD. Source data are provided as a Source Data file.

a single GxE interaction hit, where A39 loss is buffered in the presence of antimycin, suggesting a potential role of A39 in OXPHOS. We first validated the screen findings and showed that the KO's fitness defect was buffered in antimycin (Fig. 6a) and the fitness defect was enhanced in the galactose condition (Supplementary Fig. 5b), consistent with a putative role of A39 in antimycin-sensitive mitochondrial respiration. Indeed, the A39 KO had lower basal and maximum respiration (Fig. 6b), and an increased ECAR due to a compensatory upregulated glycolysis (Fig. 6c). In addition, cellular polar metabolite profiling in A39 KO cells highlighted increased serine and decreased aspartate levels (Fig. 6d, Supplementary Fig. 5d and Supplementary Data 8), a metabolic signature of OXPHOS inhibition[52–54]. Western blot analysis of A39 KO cells revealed reduced OXPHOS complex subunits suggesting a direct requirement of mitochondrial glutathione in subunit assembly and stability (Fig. 6e). The fitness defect, decreased respiration, increased glycolysis, cellular metabolic perturbations and the reduced OXPHOS complex subunit level can be rescued by re-expressing wild type A39 (Supplementary Fig. 5a, c–e and Fig. 6d, e), but cannot be rescued by the solute-binding defective mutant A39$^{D226A}$ (Supplementary Fig. 5e), supporting a direct role of GSH uptake in mitochondrial

OXPHOS subunit stability. Notably, defective mitochondrial iron uptake in the A37 KO cells also led to a similar reduced OXPHOS complex subunit levels (Fig.6f). And, the level of OXPHOS subunits is not further reduced upon additional A39 loss, a finding consistent with the buffering interaction between A39 and A37 revealed through in our CRISPR screen. Collectively, our follow-up studies support a functional interaction between A37-mediated mitochondrial iron uptake and A39-mediated mitochondrial glutathione regulation in supporting OXPHOS complex assembly.

## Discussion

The study on metabolite translocation across membrane-bound organelles via protein transporters and its impact on cellular physiology has gained momentum recently in the field of cellular metabolism. In this study, we have generated a combinatorial CRISPR library targeting the human SLC25 transporters, the largest known protein family for mitochondrial metabolite transport, with the goal to explore pairwise genetic interactions and their dependence on metabolic environment. The functional redundancy of the SLC25 transporters and conditional essentiality underscore the need for a combinatorial approach, as has

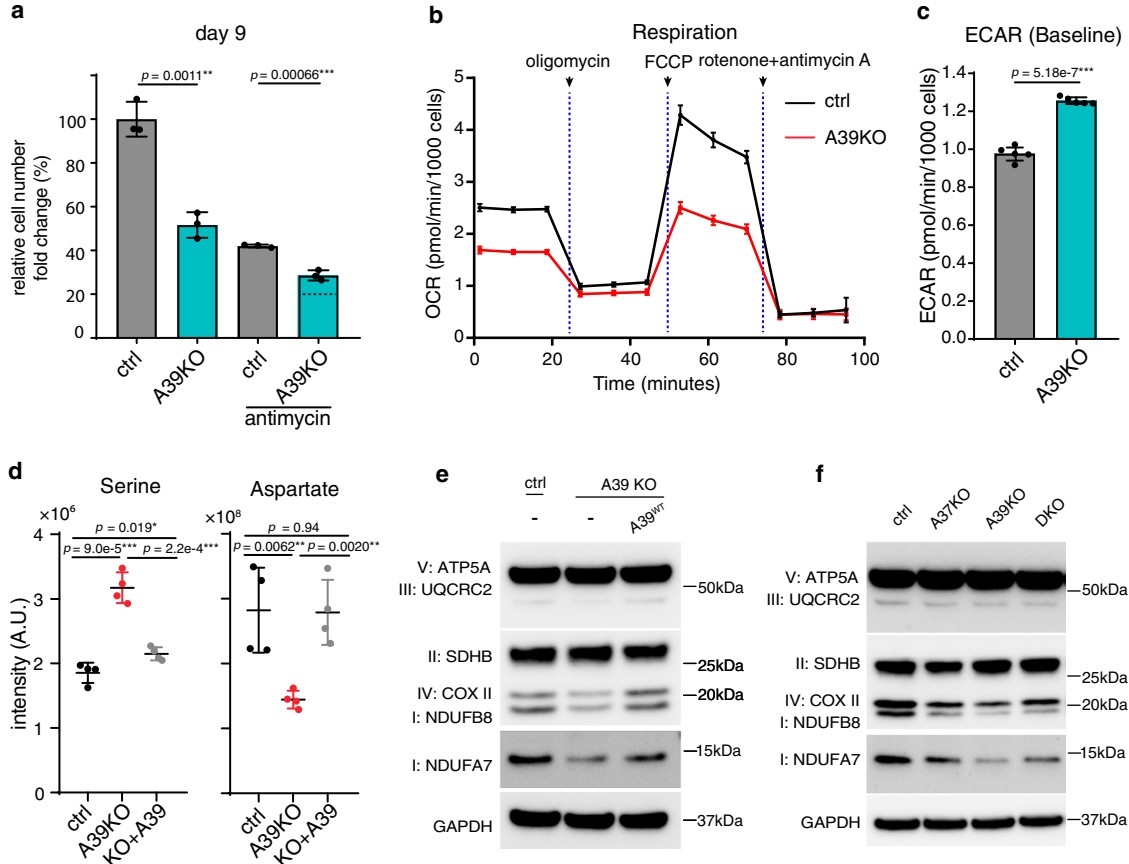

**Fig. 6 SLC25A39 converges with SLC25A37-mediated iron homeostasis to support mitochondrial OXPHOS. a** Growth fitness defect in A39 CRISPR KO cells is buffered in the antimycin condition. The dotted line indicates the expected cell number, based on an additive model, if there were no specific genetic interaction ($n = 3$). **b** Oxygen consumption rate (OCR) for A39 KO cells during Seahorse mito stress test ($n = 5$). **c** The basal extracellular acidification rate (ECAR) for A39 KO cells ($n = 5$). **d** LC-MS measurement of serine and aspartate in control, A39 KO and the rescued cells ($n = 4$). **e** Western blot for subunits of mitochondrial OXPHOS complexes in the ctrl, A39KO, and A39KO K562 cells expressing *SLC25A39* cDNA. GAPDH was used as the loading control. **f** Western blot for subunits of mitochondrial OXPHOS complexes in the ctrl, SLC25A37KO, SLC25A39KO, and DKO K562 cells. GAPDH was used as the loading control. Statistical significance was calculated using two-tailed $t$ test. Significance level were indicated as *** $p < 0.001$, ** $p < 0.01$, * $p < 0.05$ and n.s. $p > 0.05$. Data are expressed as mean ± SD. Source data are provided as a Source Data file.

been demonstrated for other SLC families[55] and the need to explore different nutrient conditions[56,57]. By screening across four different media conditions, we both recovered known interactions and discovered new insights into mitochondrial micronutrient and cofactor biology. We found that a defect in mitochondrial folate transport can be buffered by decreasing cytosolic carbohydrate metabolism. We then extended the result to an ExE interaction by showing that the cell fitness defect by methotrexate treatment can be buffered in low glucose conditions – a result predicts that the efficacy of methotrexate in humans may depend on systemic glucose homeostasis and nutritional status.

A key finding of the current study is the functional characterization of the mitochondrial transporter SLC25A39. SLC25A39 is a mitochondrial transport of unknown function, and is both a GxE and GxGxE hit, in which A39 loss is buffered by antimycin and buffered by SLC25A37 loss. Mitochondrial metabolite profiling, organelle-based uptake assay and structure-function analysis identified a critical role of A39 in regulating mitochondrial glutathione uptake. The reduced glutathione, GSH, is a tripeptide and exclusively synthesized in the cytosol by two step enzymes GCLC/GCLM and GSS (Fig. 4c)[58]. Mitochondria lack the GSH biosynthesis pathway and depend on protein transporters for its uptake. Mitochondrial glutathione, mainly

found in the reduced form, represents 10–15% of total cellular glutathione pool, and at a mM concentration similar to that of cytosol[58].

With a reactive thiol, mitochondrial GSH is critical to support intramitochondrial antioxidant defense system, redox signaling and cofactor metabolism including iron-sulfur cluster biosynthesis. The genetic interaction screen and the follow-up characterization further suggest that the impaired mitochondrial GSH uptake via A39 loss and the impaired mitochondrial iron uptake via A37 loss impact the same pathway regulating OXPHOS. Further supporting a role of A39 in iron homeostasis, we have previously identified that A37 and A39 are strongly co-expressed and co-regulate heme biosynthesis[41]. Notably, Wang et al.[14] independently reported the characterization of the role of SLC25A39 in mitochondrial glutathione import and a secondary defect of iron-sulfur cluster deficiency in cells depleted of mitochondrial glutathione. Because iron-sulfur cluster is a critical cofactor for OXPHOS subunits, defective iron-sulfur cluster biosynthesis might explain the decreased mitochondrial respiration and the reduced OXPHOS subunits in the A39 KO cells, and the buffering interaction between A39 and the iron transporter A37 loss in our screen[13]. Together, our work and that of Wang et al.[14] independently establish a role of SLC25A39 in mitochondrial glutathione import, enabling new hypotheses for future

basic science and translational studies with implications for a host of human diseases that oxidative stress is involved.

## Methods

**Cell culture.** K562 cells (CCL-243),Hela cells, HEK293T and HEK293FT cells were obtained from the ATCC and Thermo Fisher, and maintained in DMEM (Thermo Fisher, 11995) + 10% FBS (Sigma, F2442) + 100 U /ml Pen/Strep (Thermo Fisher, 15140122). The cells were tested for mycoplasma periodically. The four media conditions used in the screen and in the follow-up experiments were prepared with following reagents: DMEM (Thermo Fisher, 11966), glucose (Thermo Fisher, A2494001), galactose (Teknova, G0505), dialyzed FBS (dFBS, Thermo Fisher, 26400044), sodium pyruvate (Thermo Fisher, 11360070), and uridine (Sigma, U3750). The four media conditions are:

(1) Glc: 25 mM glucose, 10% dFBS, 1 mM sodium pyruvate, 50 µg/ml uridine, and 100 U/ml Pen/Strep
(2) Gal: 25 mM galactose, 10% dFBS, 1 mM sodium pyruvate, 50 µg/ml uridine, and 100 U/ml Pen/Strep
(3) Antimycin: 25 mM glucose, 10% dFBS, 1 mM sodium pyruvate, 50 µg/ml uridine, and 100 U/ml Pen/Strep with 100 nM antimycin (Sigma, A8674)
(4) –Pyruvate: 25 mM glucose, 10% dFBS, 50 µg/ml uridine, and 100 U/ml Pen/Strep

For folate experiment, RPMI1640 media lacking glutamine, glucose, vitamin B[12], folic acid, phenol red (US Biological, R9001–02) were further supplemented with 300 mg/l glutamine (Thermo Fisher, 25030081), 25 mM glucose, 0.005 mg/l vitamin B[12] (Sigma, V6629), 10% dFBS, 50 µg/ml uridine, 100 U/ml Pen/Strep and either 10 µM (high folate media) or 0.08 µM (low folate media) folic acid (Sigma, F8758). Methotrexate (Sigma, A6770) was used in high folate media to inhibit the folate pathway.

For SLC25A19 validation, the cells were treated with 0.1% DMSO, 100 nM Antimycin, 10 nM piericidin (Enzo Life Sciences, ALX-380-235-M002), or 10 nM oligomycin (Agilent Technologies, 103015-100), respectively.

For SLC25A39 characterization, both RPMI (Gibco, 11875) -based and DMEM (Gibco, 11995)-based media are used interchangeably with little difference. The cellular metabolite profiling results in Fig. 6d and Supplementary Fig. 5d were shown for the RPMI condition. For K562 GSH rescue experiment, K562 cells were treated with 10 mg/l reduced glutathione (Sigma, G6013).

Lentivirus for the follow-up experiments was produced using HEK293FT cells. Specifically, $1 \times 10^6$ HEK293FT cells were plated in 6 cm dish one day before induction. A mixture with 1 µg VSV-G (Addgene, 8454), 1 µg psPAX2 (Addgene, 12260), 2 µg lentiviral plasmids and 10 µl dH$_2$O was further added with 150 µl serum free DMEM and 9 µl X-tremeGENE HP DNA Transfection Reagent (Sigma, 6366244001), and incubated for 15–30 min at room temperature before adding drop-wise to the cells. Two days after induction, the spent media was collected and filtered through 0.45 µm sterile filter, and the resulting virus can be stored at -80 °C till use.

**Combinatorial CRISPR screen vectors.** The vector used for combinatorial CRISPR screens has been reported before[15] and is available through Addgene: pPapi (also known as pXPR_207). In this vector, U6 and H1 promoters express two sgRNAs; short EF1a promoter (EFS) expresses SaCas9-2A-PuromycinR (Addgene, 96921).

**pPapi-SLC25 library construction and production.** To construct the pPapi-SLC25 library, SpCas9 and SaCas9 guides were designed using the GPP sgRNA designer (https://broad.io/gpp-sgrna-design) with a quota of four guides per gene per enzyme. 53 SLC25 family members and 7 additional SLC proteins present in the Mitocarta 2.0[16] were included, as well as two guides per gene for BCL2L1, MCL1 and EEF2 as positive controls; 8 different non-cutting controls and 19 cutting controls that target olfactory receptors (ORs) across different chromosomes served as negative controls.

Pooled libraries for expression of single sgRNAs were made as previously described[15], with oligonucleotide pools obtained from CustomArray16. For cloning of Big Papi pools, oligonucleotide inserts (Ultramers, IDT) were designed with 5′ BsmBI sites followed by 20 or 21 nt crRNA, 82 nt tracrRNA, 6 nt barcode, and a 17 nt complementary sequence. The oligonucleotides for SpCas9 sgRNAs and SaCas9 sgRNAs were separately mixed together at a concentration of 5 µM each. 10 µl of each pool of oligonucleotides was then combined in a 100 µl reaction and extended using NEBNext (New England Biolabs) with an annealing temperature of 48 °C. The resulting dsDNA was purified by spin-column then ligated into the BsmBI-digested pPapi vector using 100 cycles of Golden Gate assembly with 100 ng insert and 500 ng vector using Esp3I and T7 ligase. The DNA was isopropanol precipitated and electroporated into STBL4 cells. A zero-generation (G0) plasmid DNA pool was then amplified by a second electroporation into STBL4 cells to create the G1 plasmid DNA pool, which was then used for virus production.

**Screen virus production.** For individual virus production: 24 h before transfection, HEK293T cells were seeded in 6-well dishes at a density of $1.5 \times 10^6$ cells per well in 2 ml of DMEM + 10% FBS. Transfection was performed using TransITLT1

(Mirus) transfection reagent according to the manufacturer's protocol. In brief, one solution of Opti-MEM (Corning, 66.25 µl) and LT1 (8.75 µl) was combined with a DNA mixture of the packaging plasmid pCMV_VSVG (Addgene, 8454, 250 ng), psPAX2 (Addgene, 12260, 1250 ng), and the sgRNA containing vector (e.g., pPapi, 1,250 ng). The two solutions were incubated at room temperature for 20–30 min, during which time the HEK293T cells were replenished with fresh media. After this incubation, the transfection mixture was added dropwise to the surface of the HEK293T cells, and the plates were centrifuged at $1000 \times g$ for 30 min. Following centrifugation, plates were transferred to a 37 °C incubator for 6–8 h, then the media was removed and replaced with media supplemented with 1% BSA. A larger-scale procedure was used for production of the sgRNA library; 24 h before transfection, $18 \times 10^6$ HEK293T cells were seeded in a 175 cm$^2$ tissue culture flask, with transfection performed as described above using 6 ml of Opti-MEM and 300 µl of LT1. Flasks were transferred to a 37 °C incubator for 6–8 h, then the media was aspirated and replaced with BSA-supplemented media. Virus was harvested 36 h after this media change.

**Genomic DNA (gDNA) preparation.** At least $7.5 \times 10^7$ cells from the screen were washed with PBS, and the gDNA was first isolated using Blood & cell culture DNA maxi kit (Qiagen, 13362) as per the manufacturer's instructions. The gDNA precipitant was suspended in 600 µl H$_2$O overnight, and subject to chloroform purification. Specifically, the suspended DNA was mixed with 600 µl phenol:chloroform:isoamyl alcohol (25:24:1) (Sigma, P2069-100ml) in the fume hood by inverting the tube. The tube was further incubated at room temperature for 5 min, and spun at highest speed at room temperature using table top centrifuge for 15 min. The top aqueous phase (~540 µl) was collected, mixed with 60 µl NaAcetate (3 M pH 5.5), 2 µl glycogen (20 µg/µl) and 600 µl isopropanol, before frozen at –80 °C for at least one hour. After thawing, the DNA pellet was washed with 1 ml cold 70% ethanol, air-dried, and suspended in 600 µl ultrapure H$_2$O overnight. The gDNA was quantified using Qubit dsDNA HS Assay Kit (Thermo Fischer).

**PCR of the sgRNA regions, NGS and deconvolution.** For the pPapi vector, dual sgRNA cassettes and plasmid DNA (pDNA) were PCR-amplified and barcoded with sequencing adaptors using ExTaq DNA Polymerase (Clontech) following the same procedure as in Najm et al. 2018 (Ref. [15].) except for the sequencing primer. The primer sequence used here uses a shorter read length and includes the super T modified bases to stabilize the duplex: CAACTTGAAAAAGTGGCACCGAGTC GGTGC T/iSuper-dT/TT/iSuper-dT/T. This primer binds immediately upstream of the 6nt barcode for the SpCas9 guide, then reads the constant overlap extension region, and finally the 6 nt barcode for the SaCas9 guide.

Specifically, amplified samples were then purified with Agencourt AMPure XP SPRI beads (Beckman Coulter, A63880) according to manufacturer's instructions and sequenced on a NextSeq sequencer (Illumina) with 50 nt single-end reads, with a 10% spike-in of PhiX DNA. Deconvolution of single sgRNA expressing vectors was as described before[15]. For the pPapi vector, reads of the first sgRNA were counted by first searching in the sequencing read for CACCG, the part of the vector sequence that immediately precedes the 20-nucleotide U6 promoter-driven SpCas9 sgRNA. The sgRNA sequence following this search string was mapped to a reference file with all sgRNAs in the library. To find the H1 promoter-driven SaCas9 sgRNA, two 21-nucleotide sequences were compared: the sequence beginning 194 nucleotides after the SpCas9 sgRNA and the sequence following the S. aureus tracr sequence (CTTAAAC). If the sequences matched, the 21 nt sequence was then mapped to the reference file with all SaCas9 sgRNA. For some sequencing lanes with poorer quality, the reference file with the SaCas9 sgRNAs sequences was shortened, such that fewer than 21 nts were needed to match in order to determine the identity of the sgRNA in that position. Reads were then assigned to the appropriate experimental condition based on the 8-nucleotide P7-appended barcode.

**CRISPR screen.** Two independent replicates were performed. For each replicate, K562 cells expressing Cas9-2A-EGFP[9] were first sorted for GFP-expressing cells, and within one week after sorting, the cells were spin-infected with the pPapi-SLC25 CRISPR lentiviral library. Specifically, approximately $2–3 \times 10^8$ cells were aliquoted in 12 well plates with $3 \times 10^6$ cells in each well for up to 96 wells, and further added with virus volume corresponding to 20–30% infection efficiency and polybrene at a final concentration at 5 µg/ml. Together, each well holds 2 ml infection mixture, and all the 12 well plates were spun at $1000 \times g$ for 2 h at 37 °C. After the spin, the media were carefully removed and replaced with 2 ml fresh media, followed by incubation overnight. The infected cells were selected for 4 days with 2ug/ml puromycin (Thermo Fischer, A1113803) at a seeding density of $1 \times 10^5$ per ml in T175 flask, and further expanded in regular cell culture media for 3 days at a seeding density of $1 \times 10^5$ per ml, and this time point is considered d 0. Then for the growth fitness screen, the cells were washed with PBS, and cultured for additional 15 days (d 15) into four different media conditions mentioned above in spinner flask (Corning, CLS3561) with vent caps (Corning, CLS3567). During the screen, in order to keep the cells at the exponential growth phase, cells were passaged every three days and seeded at following density: $1 \times 10^5$ per ml for glucose, $2 \times 10^5$ per ml for galactose, $1.5 \times 10^5$ per ml for antimycin, and $1 \times 10^5$ per

ml for –pyruvate. During each passage, ~10 % of the spent media were kept and mixed with the fresh media. For each time point, approximately $7.5 \times 10^7$ cells were pelleted at 300 g for 10 min and washed with 10 ml PBS for subsequent genomic DNA preparation.

**Screen data analysis and QC.** To infer the relative growth fitness phenotypes, we quantified and log10-normalized sgRNA plasmid abundance at day 15 in respective media conditions, and compared to the starting abundance (plasmid DNA). The raw data consisted of the read counts of 273 guides built into dual-guide plasmids, giving $273 \times 273 = 74{,}529$ total guide-guide pairs. The log-fold-change of the raw data was calculated:

Log fold change (LFC) = log2(frequency_condition×1000000 + 1) – log2 (frequency_pDNA×1000000 + 1)

The following filtering steps were performed to decrease noise and increase power of detection (see Supplementary Fig. 1c):

(1) Guide-guide pairs with a low read count in the pDNA were filtered using an empirical cut-off based on the read count distribution. Individual guides with median read counts lower than the empirical cut-off were also filtered (1177).
(2) Plasmids which contained a non-cutting control guide were filtered (4304).
(3) A small number of cutting control guides showed lethal single-gene phenotypes, and were subsequently filtered (1016).
(4) For each gene in each batch, across all conditions, a correlation matrix between all guide sequences was calculated. When the sum of the correlations for a guide sequence across a row was less than 0, the guide-guide pair was marked as "poor" since the guide-guide pair did not correlate with itself over multiple conditions. All unique guides marked as poor in either batch 1 ($n = 40$) or batch 2 ($n = 39$) were removed (unique $n = 50$). In total, 12,511 non-correlated guide-guide pairs were removed in this step.

The remaining 55,521 guide-guide pairs were used in subsequent analysis. It was confirmed that no gene or gene pair was unrepresented in the data. For all downstream analyses, GeneA x GeneB pair and GeneB x GeneA pair are combined.

It is worth noting that the presence of both cutting negative control sgRNA guides and non-cutting negative control sgRNA guides in our library allowed experimental comparison between the two types of negative controls. Previous genome-wide CRISPR screens suggest that double-stranded DNA cutting by Cas9 enzymes might lead to deleterious effects on cell proliferation[20], and thus control guides that cut the genome might be better negative controls than non-cutting control guides. Indeed, in our screen, the 19 cutting controls that target olfactory receptors (ORs) across different chromosomes showed less variation within and across conditions and also confer a growth phenotype very close to neutral (LFC = 0). On the other hand, the cells bearing the non-cutting control guides appear to grow better than the population mean over time (Supplementary Fig. 1f). For this reason, we only used cutting guides for downstream analysis, and removed any control guides that showed deleterious phenotypes.

After the quality control and filtering steps mentioned above, the two batches of the screen were found to correlate highly (Pearson's R = 0.86, two-tailed p value < 2.2e–16, Supplementary Fig. 1a). For each gene, there is high correlation between the SaCas9 and SpCas9 guides, as the median LFC of the Gene x Ctrl plasmids were well-correlated to the Ctrl x Gene plasmids (Pearson's R = 0.78, two-tailed p value < 2.2e–16, Supplementary Fig. 1b). Within each Cas9 position, the sgRNA guides were more correlated in the SpCas9 position (mean Pearson's R = 0.92) than in the SaCas9 position (mean Pearson's R = 0.82; two-tailed t test p value < 2.2e–16, Supplementary Fig. 1e).

**GxE interactions data analysis.** In order to assay for gene-by-environment interaction, we utilized a subset of our dataset which included only guide-guide pairs where at least one guide was a control guide, giving us single-knock-outs and the ctrl-ctrl pairs. Since growth rates differed between conditions, we took the z-score of the LFC within a condition to compare across conditions. The z-score was calculated relative to the control-control distribution in each condition, which was thought to best reflect the neutral growth phenotype within a condition. We then performed the following regression:

z-score ~ gene + gene:condition

**GxG interactions data analysis.** Gene-gene interactions were modeled using a previously-described scoring system called the pi-score:[33]

$$\left(\hat{\alpha}, \hat{\phi}_{ij}, \hat{\psi}_{kl}\right) = \arg\min \sum_{ijkl} ||C_{ijkl} - \alpha - \phi_{ij} - \psi_{kl}||_1 \quad (1)$$

$$s.t. \sum_{i \in \text{neg}} \phi_{ij} = 0 \text{ and } \sum_{k \in \text{neg}} \psi_{kl} = 0 \quad (2)$$

$$\pi_{ik} = \begin{cases} \text{median}_{jl}\left(C_{ijkl} - \hat{\alpha} - \hat{\phi}_{ij} - \hat{\psi}_{kl}\right), & i \neq k \\ 0, & \text{if } i = k \end{cases} \quad (3)$$

Where, for gene $i$ with guide $j$, and gene $k$ with guide $l$, the log fold change of the double knock outs ($C_{ijkl}$) are the data in the matrix. Alpha is an intercept term, and

phi and psi represent the single gene terms. Robust linear regression is used to estimate alpha, phi, and psi across the dataset, subject to the conditions that control genes have a distribution centered at 0. The pi-score for a double knock-out of gene $i$ with gene $k$ is the median over all guides $j$ and $l$ of the difference between the log fold change and the estimated parameters. Pi-scores are not defined as 0 for auto-double-knockouts. Significance is estimated from the guide-level data using the *limma* package to create a linear model within R. Significance values from the regression were then corrected for false discovery rate using the Benjamini-Hochberg method (the default within *limma*); it is these adjusted p-values which are reported. The effect size is also reported. All data analysis was done using custom R scripts performing standard analyses using R version 3.6.0 and Bio-conductor release 3.9.

**GxGxE interaction filtering.** Gene-gene interaction can also vary by condition. In order to look at gene-gene interaction in the context of different environments, we first restricted ourselves to all gene-gene interactions with FDR < 2% that included only SLC25 family member genes. Then we removed gene-gene interactions that met the following disqualification criteria:

(1) Small effect size (all pi-score < 0.25 & pi-score > -0.25)
(2) "Ceiling effect": The two genes had positive effect sizes as single knock-outs and a negative pi-score, indicating the possibility that the double knock-out had hit a growth ceiling.
(3) "No deader-than-dead effect": The two genes had a negative effect size (LFC < -0.75) and a positive pi-score across all conditions, can't be "deader than dead".

We were left with a curated list of 9 gene-gene pairs that show interactions of potential interest for follow-up. We applied a z-score transformation to the pi-scores to correct for differences of growth rates between conditions leading to blunted pi-scores in conditions with slower growth rates. The resulting heatmap visually shows several gene-gene pairs have GxGxE interaction.

**Hit validations.** For the follow-up experiments for single KOs, following spCas9 sgRNAs were synthesized and cloned into the lentiCRISPR v2 vector (Genscript), and the resulting plasmids were used to generate the knockout lines.

SLC25A32 #1 TAATGGGTTTGTAATGCAGA
SLC25A32 #2 GGAGTAACCCCAAATATATG
SLC25A19 #1 GGGTGCGCAGAACATCCACG
SLC25A19 #2 GCTCCTATACATGGTCCCCA
EGFP control GGGCGAGGAGCTGTTCACCG

For the DKO cells, spCas9 guides for these genes were synthesized and cloned into either lentiCRISPR v2 vector (puromycin selection) and pXPR_BRD051 (hygromycin selection) vectors, and the resulting plasmids were used to generate the single knockout (puromycin selection) and the double knockout (a combination of puromycin and hygromycin selection) lines.

SLC25A37#1 GCACTCGGTCATGTACCCGG
SLC25A37#2 AGCAATCAGCTGCATCCGGA
SLC25A39#1 TGACCTCTACGCACCCATGG
SLC25A39#2 GTGAAGATCGTGAGGCACGA
OR2M4-chr01 (OR2) CCATAAGGGACAGTTGACTG
OR11A1-chr06 (OR11) GTGATGCCAAAAATGCTGGA
SLC25A1 AGTTCCTCAGCAACCACATG
SLC25A5 CACCGGGGAGTTCTGTCCTTCTGGCG
SLC25A6 CACCGCGAAGTTGAGGGCTTGAGTG
SLC25A20 TCCAGGAGTCATGATTCCTG
SLC25A28 CACCGTAAGAACGGAGGGCCTATGG
SLC25A36 TGGTGCATCTGTTTGCCGGA
MTCH1 GGACAACGCCCCGACCACTG
MTCH2 AGCACTTTCACGTACATGAG

For the study of SLC25A39, Lentiviral plasmids were used to ectopically express human SLC25A39^WT, SLC25A39^D226A, SLC25A39^R225A/D226A and SLC25A39^K329A protein with a C-terminal FLAG tag.

For rapid mitochondrial isolation experiments, following plasmids were used: pMXs-3XHA-EGFP-OMP25 (Addgene plasmid #83356) and pMXs-3XMyc-EGFP-OMP25 (Addgene plasmid #83355).

To generate the DKOs and corresponding controls, cells were first infected with plasmids bearing the hygromycin selection cassette followed by selection with 0.25 mg/ml hygromycin (Roche, 10843555001) for one week, and then infected again with plasmids bearing puromycin selection cassette following by 2 μg/ml puromycin selection for two days. Two cutting controls were used here to control for the growth fitness defects caused by double-strand DNA breaks during the Cas9 cutting.

**mRNA assays.** The mRNAs were isolated from 0.5 million cells using RNeasy Mini kit (Qiagen, 74104); the cDNA was prepared using SuperScript III First-Strand Synthesis SuperMix for qRT-PCR kit (Invitrogen, 11752050). qPCR experiment was performed on the QuantStudio 6 Flex Real-Time PCR Systems using the recommended Taqman probes.

**Respiration assay**. Cellular respiration assays were performed using a seahorse XFe24 Analyzer (Agilent Technologies) coupled with BioTek Cytation 1 Cell Imager. 100k cells were plated in each well into a seahorse XFe24 culture plate that was pre-coated with cell-tak (Corning, 354240) working solution. Cellular OCR and ECAR were analyzed using the Mito Stress Test Kit (Agilent Technologies, 103015) at the following drug concentrations: 1.5 μM oligomycin, 1 μM FCCP and 0.5 μM mixture of rotenone and antimycin. At the end of the assay, the cells were stained with Hoechst dye (Thermo Fisher, 62249), and imaged with Cytation 1 for normalization.

**LC-MS methods**. LC/MS-based analyses were performed on a Q Exactive plus benchtop orbitrap mass spectrometer equipped with an Ion Max source and a HESI II probe, which was coupled to a Dionex UltiMate 3000 UPLC system (Thermo Fisher Scientific) or a QE plus coupled to a Vanquish UHPLC.

Polar metabolite detection method was adapted with minor modification from previously literature[59,60]. Cells were seeded at 1 million per well in 6 well plates for 24 hr prior to harvesting. To prepare cellular metabolite extracts, transferred the cells and medium to 2 ml tube, centrifuged at 600 × g for 3 min at 4 °C, then aspirated the spent media and washed once with 1 ml ice-cold PBS before lysis with acetonitrile-methanol-water (27:9:1 vol/vol/vol). For spent media metabolites, 50 μl spent media sample was aliquoted in the Eppendorf tube and mixed with 450 μL ice-cold acetonitrile-methanol (75:25 vol/vol). The extraction mixture was vortexed and centrifuged at 19,000 × g for 20 min at 4 °C, and 150 μl supernatant was transferred into LC-MS glass vial for analysis.

Polar metabolites were analyzed on Xbrige BEH Amide XP HILIC Column, 100 Å, 2.5 μm, 2.1 mm×100 mm (Waters, 186006091) for chromatographic separation. The column oven temperature was 27 °C and autosampler was 4 °C. Mobile phase A: 5% acetonitrile, 20 mM ammonium acetate/ammonium hydroxide, pH 9, and mobile phase B: 100% acetonitrile. LC gradient conditions at flow rate of 0.220 ml/min as following: 0 min: 85% B, 0.5 min: 85% B, 9 min: 35% B, 11 min: 2% B, 13.5 min: 85% B, 20 min: 85% B.

The polar metabolites from mitochondria were analyzed on SeQuant ZIC-pHILIC polymeric 5 μm, 150×2.1 mm column (EMD-Millipore 150460). The LC-MS condition was same as described in nucleotides section.

The mass data were acquired in the polarity switching mode with full scan mode in a range of 70–1000 m/z, with the resolution at 70,000, the AGC target at 1e[6], and the maximum injection time at 80 ms, the sheath gas flow at 50 units, the auxiliary gas flow at 10 units, the sweep gas flow at 2 units, the spray voltage at 2.5 kV, the capillary temperature at 310 °C, and the auxiliary gas heater temperature at 370 °C.

Compound discovery (Thermo Fisher Scientific) was used to pick peaks and integrate intensity from raw data. One filter was applied: max coefficient of variation within quality control samples (stdev/mean) < 0.3. The filtered metabolite lists were then annotated by searching against in-house chemical standard library with 5-ppm mass accuracy and 0.5 min retention time windows followed by manual curation. For the cellular profiling in the Supplementary Fig. 5d, consistently and significantly changed metabolites with p-value < 0.05 that can be rescued are shown.

Folate species detection method was adapted with minor modification from previously literature[31,61]. Specifically, ~2.5 × 10[6] cells were pelleted at room temperature, without washing, and extracted with dry ice cold 80% methanol containing 25 mM sodium ascorbate, 10 mM ammonium acetate. Cell extracts were vortexed and incubated on wet ice for 20 min before centrifugation at 21,000 × g for 20 min at 4 °C. Supernatants were dried down under nitrogen, re-suspended in 300 μl re-suspension buffer (0.5% (w/v) ascorbic acid + 1% (w/v) K₂HPO₄ + 0.5% 2-mercaptoethanol). The polyglutamate tails of the folate species were cleaved by incubating the extracts with 30 μl charcoal deactivated rat serum at 37 °C for 1.5 h and adjusted to pH 4 with 100 μl of 1 % formic acid. The charcoal deactivated rat serum was prepared by incubating 5 ml rat serum (Sigma, R9759) with 250 mg activated charcoal (Sigma, 9157) followed by end-to-end rotation at room temperature for 3 h, the resulting serum were cleared by centrifugation and passing through 0.2 μm filters. The treated samples were cleaned up using Agilent Bond Elute 96 square well plate (Agilent Technologies, A3961550). SPE wells were conditioned with 300 μl 0.25 % (w/v) ammonium acetate in 90% methanol, equilibrated with 300 μl 0.25% ammonium acetate before sample loading, washed with 300 μl 0.25% ammonium acetate and eluted with 300 μl 0.25 % ammonium acetate + 0.5 % mercaptoethanol in 50% methanol. Samples were dried down using nitrogen dryer, re-suspended in 75 μl water before analysis, and 10 μl sample were analyzed by LC-MS.

Folate species were analyzed on ACQUITY UPLC HSS T3 Column, 100 Å, 1.8 μm, 2.1 mm × 100 mm (Waters, 186003539) for chromatographic separation. The column oven temperature was 30 °C and autosampler was 4 °C. Mobile phase A: 20 mM ammonium formate, pH 5 (adjusted with formic acid), and mobile phase B: methanol. LC gradient conditions at flow rate of 0.250 ml/min as following: 0 min: 5% B, 3 min: 5% B, 11.4 min: 36% B, 11.5 min: 95% B, 14.0 min: 95% B, 14.1 min: 5% B, 20 min: 5% B.

The MS data were acquired in positive polarity in range of: 438–450 m/z, 452–462 m/z, 470–478 m/z, resolution 70,000, AGC target 5e[6] and maximum injection time of 300 ms. MS parameters were: sheath gas flow 50, aux gas flow 10, sweep gas flow 2, spray voltage 2.50, Capillary temperature 310 °C, S-lens RF level

-50 and aux gas heater temperature 370 °C. Data acquisition was done using Xcalibur 4.1 (Thermo Scientific) and data analysis was done using the Tracefinder 4.1. with 5 ppm mass tolerance. Retention time and MS/MS of folate species were matched with reference standards (supplementary Fig. 2j). The standards are folic acid (Sigma, F8758), THF (Schircks Laboratories, 16.207), 5,10-me⁺-THF (Schircks Laboratories, 16.230), 5,10-me-THF (Schircks Laboratories, 16.226), 5-methyl-THF (Schircks Laboratories, 16.235), 5-formyl-THF standards generated from (6 R,S)-5-formyl-5,6,7,8-tetrahydropteroylpenta-γ-L-glutamic acid, lithium salt (Schircks Laboratories, 16.285).

10-formyl-THF standard was synthesized as reported:[62] 200 μl 25 mM 5,10-me⁺-THF in water was added with 1 μl 2-mercaptoethanol and 5 μl 2 N KOH to a final ~ pH 8.5, and incubated at room temperature for 30 min. As 10-formyl-THF is very unstable, the synthesized standard was either analyzed immediately or stored at -80 °C. Comparing to the MS/MS spectrum of the 10-formyl-THF standard, the abovementioned extraction procedure might not preserve 10-formyl-THF in the cellular extracts.

Purine intermediates and nucleotides detection was performed as previously described[60,63]. Specifically, for purine intermediates, approximately 2×10[6] cells were seeded at a density of 5×10[5] cells/ml in corresponding media conditions and incubated overnight. The next day, equal number of cells in each condition were pelleted at 600 × g for 5 min at room temperature, without washing, and extracted with 0.5 ml dry ice-cold 80% methanol by vortex and incubation on wet ice for 20 min. The extracts were centrifuged at 21,000 × g for 20 min at 4 °C, and the supernatants were added with 1 ml dH₂O and dried down overnight by lyophilization. Metabolite extracts were suspended in 100 μl of 60 % acetonitrile, vortexed, and cleaned by centrifugation at 21,000 × g for 10 min at 4 °C. For NAD(P)⁺ and NAD(P)H quantification, cell pellets were lysed directly in 200 μl ice-cold acetonitrile-methanol-water (4:4:2) containing 0.1 M formic acid containing 1 μM of ¹³C₅ NAD + and 1 μM of d5-NADH as internal standards. Wash steps were omitted to avoid hydride exchange. Lysates were immediately neutralized with 17.5 μl of 15% (w/v) ammonium bicarbonate solution (after 2 min of lysis buffer addition), incubated on ice for 20 min and centrifuged at 21,000 × g for 20 min at 4 °C. Supernatants were analyzed on the same day by the ZIC-pHILIC method mentioned below. Calibration curves of NAD⁺, NADH, NADP⁺ and NADPH were generated in extraction solvent. External calibration curve was used for the NADP⁺ and NADPH quantification because of lack of respective isotope labeled standards.

Nucleotides were separated by SeQuant ZIC-pHILIC polymeric 5 μm, 150×2.1 mm column (EMD-Millipore 150460). Mobile phase A: 20 mM ammonium carbonate in water, pH 9.6 (adjusted with ammonium hydroxide), and mobile phase B: acetonitrile. The column was held at 27 °C, injection volume 5 μl, and an autosampler temperature of 4 °C. LC condition at flow rate of 0.15 ml/min as following: 0 min: 80% B, 0.5 min: 80% B, 20.5 min: 20% B, 21.3 min: 20% B, 21.5 min: 80% B till 29 min. MS data were acquired with polarity switching mode. MS parameters were: sheath gas flow 30, aux gas flow 7, sweep gas flow 2, spray voltage 2.80 for negative and 3.80 for positive, capillary temperature 310 °C, S-lens RF level 50 and aux gas heater temp 370 °C. Data acquisition was performed using Xcalibur 4.1 (Thermo Scientific) and performed in full scan mode with a range of 70–1000 m/z, resolution 70,000, AGC target 1e[6] and maximum injection time of 80 ms. Data analysis was done using Xcalibur 4.1 (Thermo Scientific) and the quality of integration for each chromatographic peak was reviewed. Metabolite annotation was done base on accurate mass (±5 ppm) and matching retention time as well as MS/MS fragmentation pattern from the pooled QC sample against in-house retention time + MS/MS library of reference standards. Metabolites which have CV < 20% in pooled QC were used for the statistical analysis.

**Sequence analysis and structural modeling**. SLC25A39 orthologs in eukaryotic organisms were identified as Bidirectional best hits (BBH) from either protein BLAST or tblastn search. The sequences were aligned using MUSCLE (MUltiple Sequence Comparison by Log-Expectation) and the secondary structure annotated using ESPript.

A homology model of human SLC25A39 was predicted using the SWISS-MODEL server based on the ADP/ATP ANT transporter c-state conformation (PDB: 1OKC). The putative substrate binding residues were predicted based on the previous structural and structure-function analysis[49,50,51].

**Mitochondrial isolation and uptake assay**. A rapid mitochondria immunoisolation method was adopted as previously described[47,48]. Specifically, approximately 30 million cell lines expressing mitoIP constructs were washed with PBS and collected at 1,000 × g for 2 min at 4 °C, suspended in 1 ml KPBS buffer (136 mM KCl, 10 mM KH₂PO₄, pH 7.25) to homogenous single cell suspension, and lysed by 20 gentle strokes in the 2 ml hand-held Dounce tissue grinder with tight-fitting pestle B (VMR, Kontes). Cell lysates were cleared by centrifugation at 1000 × g for 2 min at 4 °C, and transferred to a new 1.5 ml EP tube containing 100 μl pre-washed anti-HA magnetic beads (Pierce, 88837). After an end-to-end rotator incubation for 3.5 min in the cold room, the isolated mitochondria were washed with KPBS buffer for 3 times. For the metabolite profiling experiment, mitochondrial metabolites were extracted with 50 μl acetonitrile-methanol-water (27:9:1 vol/vol/vol). For western blotting experiment, mitochondrial proteins were lysed with 100 μl RIPA lysis buffer.

For mitochondrial uptake assay, the isolated mitochondria were incubated with 0.2 mM labeled GSH, GSH-[glycine-13C2, 15 N] (Sigma, 683620), in 100 μl assay buffer (110 mM sucrose, 20 mM HEPES pH7.4, 10 mM $KH_2PO_4$, 3 mM $MgCl_2$, 1 mM EGTA, 0.1% BSA). The uptake assay is performed as room temperature by a horizontal shaking at 170 rpm and manually mix every 10 min. At indicated time point, an aliquot of isolated mitochondria suspension was taken and washed three times with KPBS buffer, and metabolite extracted with 50 μl ice acetonitrile-methanol-water (27:9:1 vol/vol/vol) for immediate LC-MS analysis.

**Western blotting**. Following antibodies and concentration were used: Anti-SLC25A39 (Proteintech, 14963-1-AP, 1:100); Anti-VDAC (Cell Signaling Technology, 4661 T, 1:1000); Anti-SHMT2 (Sigma-Aldrich, HPA-020549, 1:1000); Anti-Calreticulin (Cell Signaling Technology, 12238 T, 1:1000); Anti-LAMP2 (Santa Cruz Biotechnology, sc-18822, 1:1000); Anti-NDUFA7 (Proteintech, 15300-1-AP, 1:500); total OXPHOS human WB antibody Cocktail (Abcam, ab110411, 1:1000); GAPDH (Invitrogen, 39-8600, 1:1000).

**Statistics and reproducibility**. For cell characterization and metabolite profiling experiments, data are shown as mean ± s.d., $n \geq 3$ biologically independent samples unless stated otherwise. Statistical significance was calculated using two-tailed $t$ test. Significance level were indicated as *** $p < 0.001$, ** $p < 0.01$, * $p < 0.05$ and n.s. $p > 0.05$. The screens were performed in two batches and the results were validated in the follow-up studies. All the other experiments were validated at least two times. Statistical analysis was performed using GraphPad Prism 7.0a, or as reported by the relevant computational tools.

**Reporting summary**. Further information on research design is available in the Nature Research Reporting Summary linked to this article.

## Data availability
The data that support this study are available from the corresponding author upon reasonable request. The screen data and metabolomics data are available within the supplementary data. Source data are provided with this paper.

## Code availability
The code for analyzing the screening data are available as a public github repository at https://github.com/BrynMarieR/combinatorial_gxgxe_screen.

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

## Acknowledgements

We thank Yale School Medicine, Yale West Campus, Systems Biology Institute and Cancer Biology Institute for the instrumentation supports; Li Chen, Xincheng Wu and Josh Rabinowitz laboratory for advice on folate species detection; Broad Genetic Perturbation Platform for assisting the CRISPR screen; Jie Liu for technical support; Guangwei Du for advice on mitochondrial isolation; and members of the Shen lab and Mootha lab, and colleagues at Yale School of Medicine and Yale West Campus for fruitful discussions and feedback. H.S. is supported by Yale School of Medicine startup fund, NIH grants K99/R00 GM124296, Chan Zuckerberg Initiative Donor-Advised Fund, an advised fund of the Silicon Valley Community Foundation (grant 2020–221912). H.S. is also supported by Klingenstein-Simons fellowship Awards in Neuroscience and 1907 Foundation Trailblazer Award. V.K.M. is supported by NIH R35GM122455 and is an Investigator of the Howard Hughes Medical Institute.

## Author contributions

H.Shen and V.K.M. conceived of combinatorial CRISPR genetic screening. J.G.D. designed the CRISPR library; H. Shen performed the CRISPR screen with the help from T.L.T.; B.R. performed computational analysis of the CRISPR screening results; S.E.C. aided in bioinformatics analysis; O.G. provided technical assistance; H. Shah and H.Shen performed the metabolite profiling for the folates and nucleotides; H. Shen, X.S., L.S., and K.B. performed functional and metabolite analysis of A39 KO mitochondria and cells. V.K.M. supervised the combinatorial CRISPR genetic screening. H. S. supervised the combinatorial CRISPR genetic screening and the functional follow-up studies; H.Shen. and B.R. wrote the initial manuscript with input from X.S. V.K.M. edited the manuscript. All authors reviewed and approved the manuscript.

## Competing interests

V.K.M. serves on the SAB and receives equity from Janssen Pharmaceutics and 5AM Ventures. J.G.D. consults for Foghorn Therapeutics, Maze Therapeutics, Merck, Agios, and Pfizer; J.G.D. consults for and has equity in Tango Therapeutics. The remaining authors declare no competing interests.
