## [Peer Review File · Nature Communications]

Combinatorial GxGxE CRISPR screen identifies SLC25A39 in mitochondrial glutathione transport linking iron homeostasis to OXPHOSEditorial Note: This manuscript has been previously reviewed at another journal that is not operating a transparent peer review scheme. This document only contains reviewer comments and rebuttal letters for versions considered at Nature Communications.

Reviewers' Comments:

Reviewer #1:

Remarks to the Author:

The major issues with the manuscript are still with the robustness of the combinatorial approach and small effect sizes in validation. Unfortunately, these issues are not solved. Given that the focus of the manuscript is to identify genetic interactions between mitochondrial transporters, confirming the screen results more rigorously is of utmost importance in my opinion. The authors validated only those the buffering interactions (even for these, effect sizes are extremely small so unclear how generalizable) but none of the new synthetic interactions. Several of the unexpected synthetic gene pairs should be validated by using simple double CRISPR experiments.

I understand that the authors added a follow-up for one of the transporters (SLC25A39), but the way they found this hit is through a buffering effect, for which the authors also have no explanation/mechanism. Instead, the authors simply characterize the transporter which makes it challenging to understand, what is the point of the combinatorial screen. There is a big disconnect in current form of the manuscript. For a screening paper the authors need to provide more validation and some evidence for robustness. For a focused manuscript on SLC25A39, the authors should provide more evidence for its role in any cell function and some explanation for the buffering effect with iron transporter. I do not think the paper cited addresses this point and provides an opportunity for the authors to learn sth new building upon their screen.

- The most important and interesting part of the paper is the claim to identify Gene*Gene interactions. I strongly suggest to validate a few of the scoring synthetic interactions (I understand it is difficult to validate many but at least couple are certainly required in a screening paper). The authors did not perform these experiments and ignored my initial question. SLC25A5/6 or SLCA25A1/36 double KO experiment could be performed using inducible systems. This is an opportunity to validate something from the screens. Without that, it is unclear how the screen findings translate to real experiments in its current form. Additionally, the authors should also try these pairs in more cell lines (at least one more unrelated cell line). Are these hits specific to one cell line or generalizable? A more depth analysis and follow up is required in this manuscript for it to be useful for the community.

-My question regarding SLC25A19 was also not answered experimentally but simply by citing a previous screen (a similar question was raised by another reviewer). The authors state that "In our experience, genome-wide CRISPR screens have often revealed hits that confer cell fitness advantage in a pooled format but then cannot be validated in the follow-up single gene". This is not a reason not to validate a screen result. What if the synthetic lethality screen in this manuscript has the same issue.

- Fig 6c: The western blot is very hard to interpret. Why is there a difference in size between endogenous and over expression (the protein has no isoform)? Is the protein modified when over expressed? Or the endogenous protein is not recognized properly? The bottom band seems more like a non-specific band. Otherwise, it looks like it is only 50% knock out. This needs to be rerun and corrected or explained as critical for the findings in later figures.

-The authors show that SLC25A39 loss causes an extremely small decrease in cell number. This is so miniscule that it requires much better controls than simply showing a clone for one specific cell line.

How do the authors know that the effect is not a clonal effect? The gold standard in the field is to use at least 2 different clones with corresponding cDNAs to rescue the phenotype. Simply showing one clone and comparing to parental cells is not the right control. Even better is that these results should be replicated in one more cell line? Similarly, Fig 4d: The decrease in respiration should be rescued by addition of glutathione/antioxidants and compared to correct controls.

Fig 6e: Results in Fig6e should be normalized to the amount of SLC25A39 protein in the western blots (The over expression amounts are vastly different between each other).

Reviewer #3:

Remarks to the Author:

We have also reviewed a previous version of this manuscript. The authors addressed all our comments and requests for clarifications sufficiently. Moreover, they now provide an in-depth follow-up of the SLC25A39 x SLC25A37 interaction and clearly demonstrate that the previously little studied SLC25A39 imports GSH into the mitochondria. The additional data is convincing. We have no further questions or objections.

Point-by-point response

Reviewer #1

Reviewer #1 (Remarks to the Author):

The major issues with the manuscript are still with the robustness of the combinatorial approach and small effect sizes in validation. Unfortunately, these issues are not solved. Given that the focus of the manuscript is to identify genetic interactions between mitochondrial transporters, confirming the screen results more rigorously is of utmost importance in my opinion. The authors validated only those the buffering interactions (even for these, effect sizes are extremely small so unclear how generalizable) but none of the new synthetic interactions. Several of the unexpected synthetic gene pairs should be validated by using simple double CRISPR experiments.

Thank you for the suggestions. We have now provided follow-up experiments using the single CRISPR knockout and double CRISPR knockout to validate nine environment state-dependent genetic interaction hits from the CRISPR screen. These genetic interaction hits identified from the screen were shown in the original Figure 3a. Together with the previously validated results, the validated genetic interactions now include synthetic sick interaction between SLC25A5 × SLC25A6 in galactose (new data, Figure 3c); SLC25A37 × SLC25A28 in glucose and in antimycin (new data, Supplementary Figure 3d); SLC25A36 × MTCH2 in glucose (new data, Supplementary Figure 3e); SLC25A36 × SLC25A1 in glucose (new data, Supplementary Figure 3f); MTCH1 × MTCH2 in glucose, in antimycin and in galactose (new data, Supplementary Figure 3g); MTCH2 × SLC25A20 in antimycin (new data, Supplementary Figure 3h); and buffering interaction between SLC25A37 × SLC25A39 in glucose and in antimycin (Figure 3g).

I understand that the authors added a follow-up for one of the transporters (SLC25A39), but the way they found this hit is through a buffering effect, for which the authors also have no explanation/mechanism. Instead, the authors simply characterize the transporter which makes it challenging to understand, what is the point of the combinatorial screen. There is a big disconnect in current form of the manuscript. For a screening paper the authors need to provide more validation and some evidence for robustness. For a focused manuscript on SLC25A39, the authors should provide more evidence for its role in any cell function and some explanation for the buffering effect with iron transporter. I do not think the paper cited addresses this point and provides an opportunity for the authors to learn sth new building upon their screen.

We thank the reviewer for the comments. We now performed brand-new experiments to gain mechanistic insights in the buffering interaction between SLC25A39 and SLC25A37, which was identified through the screen. We expanded the experiments on the role of SLC25A39 in OXPHOS, discovered from our previous version, and now presented new data to show that SLC25A39 loss in the K562 cells leads to a reduced mitochondrial OXPHOS subunit protein level (shown below, left, as well as in Figure 6e). This defect can only be rescued by re-expressing wild type SLC25A39, but not the mutant SLC25A39 defective in GSH binding (shown below, right, Supplementary Figure 5e).

In addition, the SLC25A39 and SLC25A37 double KO cells appear to have a similar level of reduced OXPHOS subunits as either SLC25A37 and SLC25A39 single KO cells (shown below as well as in Figure 6f), with no additive effect, supporting a buffering interaction between SLC25A37 and SLC25A39 loss in OXPHOS subunit stability. We therefore conclude a convergent role of SLC25A39-mediated GSH import and SLC25A37-mediated iron import toward supporting mitochondrial OXPHOS. This finding explains the genetic interaction identified through the CRISPR screen and invites further mechanistic study.

We would like to point out that while our manuscript is in the previous round of revision, another manuscript by Wang et al. was posted on BioRxiv and subsequently appeared in the journal Nature, which reported a similar finding that SLC25A39 is necessary for the mitochondrial glutathione level and is critical for the stability of the mitochondrial iron-sulfur containing proteins. Because iron-sulfur is the critical cofactor for OXPHOS complexes, our new report on a convergent role between A39-mediated glutathione import and A37-mediated iron import onto supporting OXPHOS provided the functional evidence that nicely complements the other study.

- The most important and interesting part of the paper is the claim to identify Gene*Gene interactions. I strongly suggest to validate a few of the scoring synthetic interactions (I understand it is difficult to validate many but at least couple are certainly required in a screening paper). The authors did not perform these experiments and ignored my initial question. SLC25A5/6 or SLCA25A1/36 double KO experiment could be performed using inducible systems. This is an opportunity to validate something from the screens. Without that, it is unclear how the screen findings translate to real

experiments in its current form. Additionally, the authors should also try these pairs in more cell lines (at least one more unrelated cell line). Are these hits specific to one cell line or generalizable? A more depth analysis and follow up is required in this manuscript for it to be useful for the community.

Thank you for the suggestions. We have now provided follow-up experiments using the single CRISPR knockout and double CRISPR knockout to validate the genetic interaction CRISPR screen hits that were shown in the Figure 3a. Together with the previous and new results, the validated genetic interactions now include synthetic sick interaction between SLC25A5 × SLC25A6 in galactose (Figure 3c); SLC25A37 × SLC25A28 in glucose and in antimycin (Supplementary Figure 3d); SLC25A36 × MTCH2 in glucose (Supplementary Figure 3e); SLC25A36 × SLC25A1 in glucose (Supplementary Figure 3f); MTCH1 × MTCH2 in glucose, in antimycin and in galactose (Supplementary Figure 3g); MTCH2 × SLC25A20 in antimycin (Supplementary Figure 3h); and buffering interaction between SLC25A37 × SLC25A39 in glucose and in antimycin (Figure 3g). Although we agree that it is important to validate our results in different cell lines, based on the time limitation and novelty of our discovery on SLC25A39, we focus on exploring the genetic interaction between SLC25A37 and SLC25A39. For the most significant finding regarding the role of SLC25A39 in supporting mitochondrial glutathione, we also validated in the Hela cells.

-My question regarding SLC25A19 was also not answered experimentally but simply by citing a previous screen (a similar question was raised by another reviewer). The authors state that "In our experience, genome-wide CRISPR screens have often revealed hits that confer cell fitness advantage in a pooled format but then cannot be validated in the follow-up single gene". This is not a reason not to validate a screen result. What if the synthetic lethality screen in this manuscript has the same issue.

We now validated the screen result on SLC25A19 by showing the growth fitness defect of the SLC25A19 KO cells compared to the control cells can be alleviated by a spectrum of mitochondrial inhibitors, including complex I inhibitor piericidin, complex III inhibitor antimycin and complex V inhibitor oligomycin. The results are now shown below as well as in the Supplementary Figure 2d.

- Fig 6c: The western blot is very hard to interpret. Why is there a difference in size between endogenous and over expression (the protein has no isoform)? Is the protein modified when over expressed? Or the endogenous protein is not recognized properly? The bottom band seems more like a non-specific band. Otherwise, it looks like it is only 50% knock out. This needs to be rerun and corrected or explained as critical for the findings in later figures.

We thank the review for the comments which prompted us to perform additional control experiments. We now generated new lines of single clonal SLC25A39 KO cells and rerun the western blot. As shown in the figure below and also in Figure 5c and Supplementary Figure 4g, the endogenous SLC25A39 is barely detectable. The shift in larger protein size for the ectopically expressed SLC25A39 is due to the C-terminal tag with a flexible linker.

Using this new clonal SLC25A39 KO cells, we now performed additional rescue experiment to demonstrate the role of SLC25A39-mediated glutathione import in supporting OXPHOS complex subunit protein level. The new data are shown below and in Supplementary Figure 5e.

Regarding the residual, endogenous SLC25A39 protein in the SLC25A39 pooled knockout cells shown in the previous version of the figure, we suspect that it might be due to an upregulation of SLC25A39 protein level upon mitochondrial glutathione depletion, as an indication of the expected phenotype. Such a striking feedback upregulation of SLC25A39 protein level upon glutathione depletion was identified by Wang et al. 2021 Nature paper while our work is in revision. We therefore generated the clonal KO cells and provided new data.

-The authors show that SLC25A39 loss causes an extremely small decrease in cell number. This is so miniscule that it requires much better controls than

simply showing a clone for one specific cell line. How do the authors know that the effect is not a clonal effect? The gold standard in the field is to use at least 2 different clones with corresponding cDNAs to rescue the phenotype. Simply showing one clone and comparing to parental cells is not the right control. Even better is that these results should be replicated in one more cell line? Similarly, Fig 4d: The decrease in respiration should be rescued by addition of glutathione/antioxidants and compared to correct controls.

We would like to draw the reviewer's attention that all the growth fitness phenotypes in the previous manuscript were plotted on the log scale that might appear milder, instead of the linear scale. In the new version of the manuscript, we now replotted the growth fitness using the linear scale, in order to better illustrate the growth fitness phenotype.

We now performed and provided new control experiments that showed two lines of SLC25A39 KO cells using different sgRNAs both exhibited growth fitness defects. The data are shown below and also in Supplementary Figure 4a. The key findings regarding a depletion of mitochondrial glutathione in the SLC25A39 KO cells were validated both using another CRISPR sgRNA guide (Supplementary Figure 4d), as well as in another cell line Hela cells (Supplementary Figure 4c).

We also performed new experiments showing that the growth defect in the SLC25A39 KO cells cannot be rescued by excess amount of GSH (10 mg/L, 10x RPMI media GSH) in the culture media (below and supplementary Figure 4i). We are confident that the growth defect in the KO cells is on-target effect because the growth defect can be rescued by re-expressing wild type SLC25A39 but not the SLC25A39 mutant defective in glutathione binding (Supplementary Figure 4h). Therefore, we concluded that cellular glutathione level is not limiting for its mitochondrial import, which is consistent with little impairment of whole cellular glutathione, in fact slightly increased, in the A39 KO cells (Figure 4d). The depletion of mitochondrial glutathione phenotype is validated in the SLC25A39 CRISPR KO K562 and Hela cells (Supplementary Figure 4c).

The new data are shown below as well as in Supplementary Figure 4i.

-Fig 6e: Results in Fig6e should be normalized to the amount of SLC25A39 protein in the western blots (The over expression amounts are vastly different between each other).

We were not able to perform the normalization because there is very little endogenous SLC25A39 detected in the SLC25A39 KO cells to serve as the normalization denominator. The ectopically expressed mutant SLC25A39 protein is comparable with the wild type SLC25A39 protein, except for K329A mutant that is expressed at a higher level, probably because that this mutant affects the conformation and therefore protein stability.

Reviewer #3

Reviewer #3 (Remarks to the Author):

We have also reviewed a previous version of this manuscript. The authors addressed all our comments and requests for clarifications sufficiently. Moreover, they now provide an in-depth follow-up of the SLC25A39 x SLC25A37 interaction and clearly demonstrate that the previously little studied SLC25A39 imports GSH into the mitochondria. The additional data is convincing. We have no further questions or objections.

Thank you.